# PEAC: Unsupervised Pre-training for Cross-Embodiment Reinforcement Learning

**Chengyang Ying**[1]  **Zhongkai Hao**[1]  **Xinning Zhou**[1]  **Xuezhou Xu**[1]
**Hang Su**[1,2*]  **Xingxing Zhang**[1]  **Jun Zhu**[1,2]

[1]Department of Computer Science & Technology, Institute for AI, BNRist Center,
Tsinghua-Bosch Joint ML Center, THBI Lab, Tsinghua University
[2]Pazhou Lab (Huangpu), Guangzhou, China
ycy21@mails.tsinghua.edu.cn

## Abstract

Designing generalizable agents capable of adapting to diverse embodiments has achieved significant attention in Reinforcement Learning (RL), which is critical for deploying RL agents in various real-world applications. Previous Cross-Embodiment RL approaches have focused on transferring knowledge across embodiments within specific tasks. These methods often result in knowledge tightly coupled with those tasks and fail to adequately capture the distinct characteristics of different embodiments. To address this limitation, we introduce the notion of Cross-Embodiment Unsupervised RL (CEURL), which leverages unsupervised learning to enable agents to acquire embodiment-aware and task-agnostic knowledge through online interactions within reward-free environments. We formulate CEURL as a novel Controlled Embodiment Markov Decision Process (CE-MDP) and systematically analyze CEURL's pre-training objectives under CE-MDP. Based on these analyses, we develop a novel algorithm Pre-trained Embodiment-Aware Control (PEAC) for handling CEURL, incorporating an intrinsic reward function specifically designed for cross-embodiment pre-training. PEAC not only provides an intuitive optimization strategy for cross-embodiment pre-training but also can integrate flexibly with existing unsupervised RL methods, facilitating cross-embodiment exploration and skill discovery. Extensive experiments in both simulated (e.g., DMC and Robosuite) and real-world environments (e.g., legged locomotion) demonstrate that PEAC significantly improves adaptation performance and cross-embodiment generalization, demonstrating its effectiveness in overcoming the unique challenges of CEURL. The project page and code are in https://yingchengyang.github.io/ceurl.

## 1 Introduction

Cross-embodiment reinforcement learning (RL) involves designing algorithms that effectively function across various physical embodiments. The fundamental goal is to enable agents to apply skills and strategies learned from some embodiments to other embodiments, which may own different physical dynamics, action-effectors, shapes, and so on [28, 70, 58, 52, 12, 69, 60]. This capability significantly enhances the generalization of RL agents, reducing the necessity for embodiment-specific training. By adeptly adapting to new and shifting embodiment, cross-embodiment RL ensures that agents maintain reliable performance in unpredictable real-world scenarios, thereby benefiting the deployment process and reducing the need for extensive data collection for each new embodiment.

---

*Corresponding author

38th Conference on Neural Information Processing Systems (NeurIPS 2024).

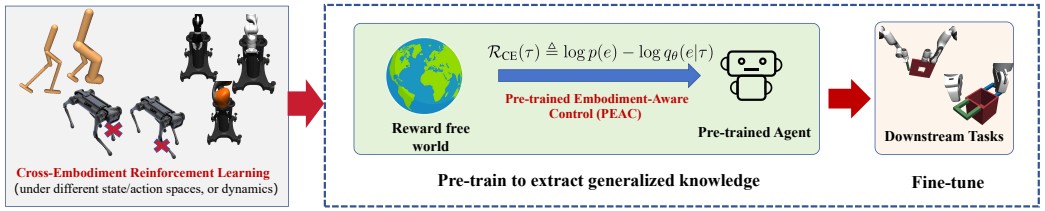

Figure 1: **Overview of Cross-Embodiment Unsupervised Reinforcement Learning (CEURL).**
The left subfigure illustrates the cross-embodiment setting with various possible embodiment changes. Directly training RL agents across embodiments under given tasks may result in task-aware rather than embodiment-aware knowledge. CEURL pre-trains agents in reward-free environments to extract embodiment-aware knowledge. The center subfigure shows the Pre-trained Embodiment-Aware Control (PEAC) algorithm, using our cross-embodiment intrinsic reward function $\mathcal{R}_{\mathrm{CE}}(\tau)$. The right subfigure demonstrates the fine-tuning phase, where pre-trained agents fast adapt to different downstream tasks, improving adaptation and generalization.

One of the primary challenges in this area is the transfer of knowledge across embodiments that have vastly different physical dynamics and environmental interactions. This requires the agent to abstract knowledge in a way that is not overly specialized to a single embodiment or some downstream tasks. However, directly training cross-embodiment agents under some given tasks will cause the learned knowledge highly related to these tasks rather than only to embodiments themselves.

Inspired by the transformative effects of unsupervised learning in natural language processing and computer vision [6, 21], which has demonstrated efficiency in extracting generalized knowledge independent of downstream tasks, we propose a natural question: *Can we pre-train cross-embodiment agents in an unsupervised manner, i.e., online cross-embodiment pre-training in reward-free environments, to capture generalized knowledge only related to embodiments?* Existing unsupervised RL techniques, including exploration [45, 38] and skill discovery [10, 26] ones, typically involve pre-training agents by engaging a single embodiment within a controlled Markov Decision Process (MDP) that lacks extrinsic reward signals. These pre-trained agents are then expected to quickly fine-tune to any downstream tasks characterized by extrinsic rewards using this specific embodiment. This approach of unsupervised RL fosters the development of policies that are not overly specialized to specific tasks or reward structures but are rather driven by intrinsic motivations of embodiments, which shows the potential for discovering more generalized knowledge across different embodiments.

In this work, we adapt the unsupervised RL paradigm to the cross-embodiment setting, introducing the concept of Cross-Embodiment Unsupervised RL (CEURL). This setting involves pre-training with a distribution of embodiments in reward-free environments, followed by fine-tuning to handle specific downstream tasks through these embodiments. These embodiments may own similar structures so that we can abstract generalized knowledge from them. To analyze CEURL and design corresponding algorithms, we formulate it as a Controlled Embodiment Markov Decision Process (CE-MDP), which comprises a distribution of controlled MDPs, each defined by its unique embodiment context. Compared to the traditional single-embodiment setting, the CE-MDP framework addresses the additional complexity caused by the inherent variability among embodiments. We then extend the information geometry analyses of the controlled MDP [11] to better explain the complexity of CE-MDP. Our findings indicate that skill vertices within CE-MDP may no longer be simple deterministic policies and the behaviors across different embodiments can display substantial variability.

To address the complexities of CE-MDP, we undertake an in-depth analysis of the pre-training objective in CE-MDP. We aim to enable our pre-trained agent to quickly fine-tune for any downstream tasks denoted as $\mathcal{R}_{\mathrm{ext}}$, especially under the worst-case reward scenarios. Thus, our pre-training objective involves minimizing across $\mathcal{R}_{\mathrm{ext}}$ while maximizing the fine-tuned policy $\pi^*$, leading to a complex min-max problem (Eq. 3). We further introduce a novel Pre-trained Embodiment-Aware Control (PEAC) algorithm to optimize this objective and handle CE-MDP, which improves the agent's robustness and adaptability across various embodiments by employing a cross-embodiment intrinsic reward $\mathcal{R}_{\mathrm{CE}}$. This reward is complemented by an embodiment discriminator, which distinguishes between different embodiments. During fine-tuning, the pre-trained policy is further enhanced under the extrinsic reward, $\mathcal{R}_{\mathrm{ext}}$ with limited timesteps. Moreover, PEAC can integrate flexibly with existing single-embodiment unsupervised RL methods to achieve cross-embodiment exploration and skill discovery, resulting in two combination algorithm examples PEAC-LBS and PEAC-DIAYN.

To verify the versatility and effectiveness of our algorithm, we extensively evaluate PEAC in both simulated and real-world environments. In simulations, we choose state-based / image-based Deep-Mind Control Suite (DMC) environments extending Unsupervised RL Benchmark (URLB) [27] and different robotic arms in Robosuite [73]. Under these settings, PEAC demonstrates superior few-shot learning ability to downstream tasks, and remarkable generalization ability to unseen embodiments, surpassing existing state-of-the-art unsupervised RL models. Besides, we have evaluated PEAC in real-world Aliengo robots by considering practical joint failure settings based on Isaacgym [37], verifying PEAC's strong adaptability on different joint failures and various real-world terrains.

In summary, the main contributions are as follows:

- We propose a novel setting CEURL to enhance agents' adaptability and generalization across diverse embodiments, and then we introduce the Pre-trained Embodiment-Aware Control (PEAC) algorithm for handling CEURL.
- We integrate PEAC with existing exploration and skill discovery techniques, designing practical methods and facilitating efficient cross-embodiment exploration and skill discovery.
- Extensive experiments show that PEAC not only excels in fast fine-tuning but also effectively generalizes across new embodiments, outperforming current SOTA unsupervised RL models.

## 2 Related Work

**Cross-Embodiment RL.** Designing generalizable agents simultaneously controlling diverse embodiments has achieved significant attention in RL. A common strategy involves using expert trajectories [70, 49, 5, 66, 60], internet-scale human videos [3, 58, 13], or offline datasets [29, 59, 8, 41] to train a generalist agent that can handle various tasks across different embodiments. However, these methods are often limited by the need for large-scale, costly datasets and the availability of expert trajectories. Additionally, the discrepancy between open-loop training and closed-loop testing may lead to distribution shifts [32], adversely affecting the final performance. An alternative line of research [28, 36, 64, 4, 66, 52, 12, 68] focuses on training general agents through online interaction across diverse environments. However, these methods treat the embodiment and task as a unified training environment and overlook the role of proprioception, i.e., the internal understanding of an agent's embodiment, which has recently proven to be beneficial for representation learning and optimization in RL [23, 15]. Thus these methods may not fully capture the intrinsic properties of different embodiments by linking knowledge to specific tasks. Emerging research suggests the potential of decoupling the training of embodiment characteristics from task execution, aiming to develop a unified cross-embodiment model. This involves unsupervised pre-training across a variety of embodiments, followed by task-aware fine-tuning, enabling a single agent to adeptly manage both roles effectively.

**Unsupervised RL.** Unsupervised RL leverages interactions with reward-free environments to extract useful knowledge, such as exploratory policies, diverse skills, or world models [17, 18]. These pre-trained models are utilized to fast adapt to downstream tasks within specific embodiments and environments. Unsupervised RL methods can be categorized into two main types: exploration and skill discovery. Exploration methods aim to maximize state coverage, typically through intrinsic rewards that encourage uncertainty [45, 7, 46, 51, 38, 40, 67] or state entropy [30, 47, 35, 34]. The resulting exploratory trajectories benefit pre-training actor-critic or world models, thereby enhancing fine-tuning efficiency [27]. Skill discovery methods focus on learning an array of distinguishable skills, often by maximizing the mutual information between states and acquired skills [10, 54, 20, 55, 25, 26, 72, 24, 61]. This approach benefits from theoretical insights into the information geometry of skill state distributions, emphasizing the importance of maximizing distances between different skills [11, 22, 42, 43, 44]. Recent efforts also explore incremental skill learning in dynamic environments [53, 31]. Unlike these methods generally focus on single embodiments, we aim to develop generalizable models capable of handling downstream tasks across a variety of embodiments.

## 3 Cross-Embodiment Unsupervised RL

In this section, we analyze the cross-embodiment RL in an unsupervised manner, which is formulated as our Controlled Embodiment MDP. Then we propose a novel algorithm PEAC to optimize CE-MDP.

## 3.1 Controlled Embodiment Markov Decision Processes

Cross-embodiment RL can be formulated by contextual MDP [19] with a distribution of Markovian decision processes (MDPs) of $\{\mathcal{M}_e\}$. Cross-embodiment RL hopes to learn shared knowledge from this distribution of MDPs, which is crucial for enhancing the adaptability or generalization of agents across embodiments. However, directly optimizing agents by online interacting with $\{\mathcal{M}_e\}$ or utilizing offline datasets sampled from $\{\mathcal{M}_e\}$ may learn knowledge not only related to these embodiments but also highly related to these task reward functions in $\{\mathcal{M}_e\}$. This phenomenon may have negative impacts on learning the general knowledge across embodiments or improving the agent's generalization ability. For example, as the agent is required to handle $\{\mathcal{M}_e\}$, it will less explore the trajectories with low rewards. These trajectories, although not optimal for the embodiment in this task, might also include embodiment knowledge and be useful for other tasks. Without extrinsic task rewards, the agent is encouraged to learn embodiment-aware and task-agnostic knowledge, which can effectively adapt to any downstream task across embodiments.

In this paper, we propose to pre-train cross-embodiment agents in reward-free environments to ensure that the agent can learn knowledge only specialized in these embodiments themselves. In other words, we introduce unsupervised RL into cross-embodiment RL as a novel setting: *cross-embodiment unsupervised RL* (CEURL). As shown in Fig. 1, in CEURL, we first pre-train a general agent by interacting with the reward-free environment through varying embodiments sampled from an unknown embodiment distribution. Given any downstream task represented by the extrinsic reward $\mathcal{R}_{\text{ext}}$, the pre-trained agent is subsequently fine-tuned to control these embodiments, and other unseen embodiments from the distribution, to complete this task within limited steps (like one-tenth of the pre-training steps). Formally, we formulate CEURL as the following controlled embodiment MDP (CE-MDP):

**Definition 3.1** (**Controlled Embodiment MDP (CE-MDP)**). *A CE-MDP includes a distribution of controlled MDPs defined as $\mathcal{M}_e^c = (\mathcal{S}_e, \mathcal{A}_e, \mathcal{P}_e, \gamma)$, where $e \sim \mathcal{E}$ and $\mathcal{E}$ represents the embodiment distribution. Each embodiment may have different state spaces $\mathcal{S}_e$ and action spaces $\mathcal{A}_e$. $\mathcal{P}_e$ : $\mathcal{S}_e \times \mathcal{A}_e \to \Delta(\mathcal{S}_e)$ denoting the transition dynamics for embodiment $e$ and $\gamma$ is the discount factor. We define the state space $\mathcal{S} = \cup_e \mathcal{S}_e$ and adopt a unified action embedding space $\mathcal{A}$ with corresponding action projectors $\phi_e : \mathcal{A} \to \mathcal{A}_e$, which can be fixed or learnable.*

Thus we can establish a unified policy $\pi : \mathcal{S} \to \Delta(\mathcal{A})$ across all embodiments. For any embodiment $e$, we sample an action $\boldsymbol{a}$ from $\pi(\cdot|\boldsymbol{s})$ for a state $\boldsymbol{s} \in \mathcal{S}_e$ and execute the projected action $\phi_e(\boldsymbol{a})$. Without loss of generality, we assume $\phi_e$ is fixed and focus our analysis on the policy $\pi$. To explain the complexities of CE-MDP with varying embodiment contexts, we extend the single-embodiment information geometry analyses [11] into our cross-embodiment setting. First, we consider the discount state distribution of $\pi$ within $\mathcal{M}_e^c$ at state $\boldsymbol{s}$ as $d_\pi^e(\boldsymbol{s}) = (1 - \gamma) \sum_{t=0}^{\infty} [\gamma^t \mathcal{P}_e(\boldsymbol{s}_t = \boldsymbol{s})]$. It is well known that the trajectory return of the state-based reward function can be computed as

$$J_{\mathcal{M}_e^c, \mathcal{R}_{\text{ext}}}(\pi) \triangleq \mathbb{E}_{\tau \sim \mathcal{M}_e^c, \pi}[\mathcal{R}_{\text{ext}}(\tau)] = \frac{1}{1 - \gamma} \mathbb{E}_{\boldsymbol{s} \sim d_\pi^e}[\mathcal{R}_{\text{ext}}(\boldsymbol{s})]. \tag{1}$$

Thus, the properties of $d_\pi^e$ are significant in determining useful initializations for downstream tasks. We consider the set $\mathcal{D}^e = \{d_\pi^e \in \Delta(\mathcal{S}) \mid \forall \pi\}$, which includes all feasible $d_\pi^e$ over the probability simplex. As shown in [11], for each $e$, $\mathcal{D}^e$ is a convex set, and any useful policy, which can be optimal for certain downstream tasks under embodiment $e$, must be a vertex of $\mathcal{D}^e$, typically corresponding to deterministic policies.

However, in the context of CEURL, the unknown embodiment context $e$ introduces partial observability [14] and significant differences in the corresponding points of the same skill across different embodiments. In CE-MDP, we consider the entire embodiment space and define $d_\pi^{\mathcal{E}}(\boldsymbol{s}) = \mathbb{E}_{e \sim \mathcal{E}}[d_\pi^e(\boldsymbol{s})]$, with $\mathcal{D}^{\mathcal{E}} = \{d_\pi^{\mathcal{E}} \in \Delta(\mathcal{S}) \mid \forall \pi\}$. The primary challenge lies in the high variability of embodiments, which complicates the process of learning a policy that generalizes well across different embodiments. We demonstrate that the vertices of $\mathcal{D}^{\mathcal{E}}$ may no longer correspond to deterministic policies, as they need to handle all embodiments in the distribution. This significantly heightens the challenge of the pre-training process in CE-MDP, making it more difficult to find useful cross-embodiment skills (proofs and discussion in Appendix A.1).

To solve CEURL under the paradigm of CE-MDP, the agent will collect reward-free trajectories $\tau = (\boldsymbol{s}_0, \boldsymbol{a}_0, \boldsymbol{s}_1, ...)$ with probability $p_{\mathcal{M}_e^c, \pi}(\tau) = \mathcal{P}_e(\boldsymbol{s}_0) \prod_{t=0} \pi(\boldsymbol{a}_t|\boldsymbol{s}_t) \mathcal{P}_e(\boldsymbol{s}_{t+1}|\boldsymbol{s}_t, \boldsymbol{a}_t)$ via some

sampled embodiments $e$ during the pre-training. These trajectories are then used in CEURL methods to design intrinsic rewards $\mathcal{R}_{\text{int}}$ for pre-training agents. During fine-tuning, we will sample several embodiments $e$ from $\mathcal{E}$ and combine $\mathcal{M}_e^c$ with a downstream task represented by extrinsic rewards $\mathcal{R}_{\text{ext}}$, and agents are required to maximize the task return over all embodiments, i.e., $\mathbb{E}_{e \sim \mathcal{E}}\left[J_{\mathcal{M}_e^c, \mathcal{R}_{\text{ext}}}(\pi)\right]$, within limited steps (like one-tenth or less of the pre-training steps).

## 3.2 Pre-trained Embodiment-Aware Control

We primarily focus on the pre-training objective of CEURL, specifically determining the optimal pre-trained policy $\pi$ for CEURL. In the fine-tuning stage, given any downstream task characterized by extrinsic reward $\mathcal{R}_{\text{ext}}$, the pre-trained policy $\pi$ will be optimized into the fine-tuned policy $\pi^*$ with *limited* steps to handle $\mathcal{R}_{\text{ext}}$ via some RL algorithms like PPO [50]. Consequently, it is widely assumed that $\pi^*$ will remain close to $\pi$ during fine-tuning due to constraints on limited interactions with the environment [11]. Our *cross-embodiment fine-tuning objective* thus combines *policy improvement* under $\mathcal{R}_{\text{ext}}$ and a *policy constraint* evaluated via KL divergence

$$\mathcal{F}(\pi, \pi^*, \mathcal{R}_{\text{ext}}, e) \triangleq \underbrace{\mathbb{E}_{p_{\mathcal{M}_e^c, \pi^*}(\tau)}[\mathcal{R}_{\text{ext}}(\tau)] - \mathbb{E}_{p_{\mathcal{M}_e^c, \pi}(\tau)}[\mathcal{R}_{\text{ext}}(\tau)]}_{\text{Policy Improvement}} - \underbrace{\beta D_{\text{KL}}(p_{\mathcal{M}_e^c, \pi^*}(\tau) \| p_{\bar{\mathcal{M}}, \pi}(\tau))}_{\text{Policy Constraint}},$$
(2)

where $\beta > 0$ is the unknown trade-off parameter related to the fine-tuning steps (when fine-tuning steps tend towards infinity, $\beta$ tends to 0 and this objective converges to the original RL objective), and $\bar{\mathcal{M}}$ represents the "average embodiment MDP" satisfying that $p_{\bar{\mathcal{M}}, \pi}(\tau) = \mathbb{E}_{e \sim \mathcal{E}}\left[p_{\mathcal{M}_e^c, \pi}(\tau)\right]$. During fine-tuning, we hope to optimize $\pi^*$ by maximizing $\mathcal{F}$, i.e., the fine-tuned result is $\max_{p_{\mathcal{M}_e^c, \pi^*}(\tau)} \mathcal{F}(\pi, \pi^*, \mathcal{R}_{\text{ext}}, e)$. As the pre-trained policy $\pi$ needs to handle *any* downstream task, we consider the worst-case extrinsic reward function across the embodiment distribution, and our *cross-embodiment pre-training objective* can be formally represented as maximizing

$$\mathcal{U}(\pi, \mathcal{E}) \triangleq \mathbb{E}_{e \sim \mathcal{E}}\left[\min_{\mathcal{R}_{\text{ext}}(\tau)} \max_{p_{\mathcal{M}_e^c, \pi^*}(\tau)} \mathcal{F}(\pi, \pi^*, \mathcal{R}_{\text{ext}}, e)\right].$$
(3)

This objective is a min-max problem that is hard to optimize. Fortunately, we can simplify it as below

**Theorem 3.2** (Proof in Appendix A.2). *The pre-training objective Eq. (3) of $(\pi, \mathcal{E})$ satisfies*

$$\mathcal{U}(\pi, \mathcal{E}) = \mathbb{E}_{e \sim \mathcal{E}}\left[-\beta D_{\text{KL}}\left(p_{\mathcal{M}_e^c, \pi}(\tau) \| p_{\bar{\mathcal{M}}, \pi}(\tau)\right)\right] = \beta \mathbb{E}_{e \sim \mathcal{E}} \mathbb{E}_{\tau \sim p_{\mathcal{M}_e^c, \pi}(\tau)}\left[\log \frac{p(e)}{p_\pi(e|\tau)}\right].$$
(4)

Here $p(e)$ and $p_\pi(e|\tau)$ are embodiment prior and posterior probabilities, respectively. This result simplifies our pre-trained objective as a form easy to calculate and optimize. Also, although $\beta$ is an unknown parameter, the optimal pre-trained policy is independent of $\beta$. Based on these analyses, we propose a novel algorithm named Pre-trained Embodiment-Aware Control (PEAC). In PEAC, we first train an embodiment discriminator $q_\theta(e|\tau)$ to approximate $p_\pi(e|\tau)$, which can learn the embodiment context via historical trajectories. For cross-embodiment pre-training, PEAC then utilizes our cross-embodiment intrinsic reward, which is defined following Eq. (4) as

$$\mathcal{R}_{\text{CE}}(\tau) \triangleq \log p(e) - \log q_\theta(e|\tau).$$
(5)

Assuming the embodiment prior $p(e)$ is fixed, $\mathcal{R}_{\text{CE}}$ encourages the agent to explore the region with low $\log q_\theta(e|\tau)$. In these trajectories, the embodiment discriminator is misled, where the agent may not have explored enough or different embodiment posteriors are similar. Thus, the embodiment discriminator can boost itself from these trajectories and learned embodiment-aware contexts that can effectively represent different embodiments, which benefit generalizing to unseen embodiments.

In practice, $\mathcal{R}_{\text{CE}}$ needs to be calculated for each state $s$ rather than the whole trajectory $\tau$, also, the embodiment discriminator needs to classify the embodiment context for every state. For RL backbones that encode historical information as the hidden state $h$ like Dreamer [17, 18, 64], we directly train $q_\theta(e|h, s)$ as the discriminator and further calculate $\mathcal{R}_{\text{CE}}$. For RL algorithms with Markovian policies like PPO [50], we encode a fixed length historical state-action pair to the hidden state $h$ and also train $q_\theta(e|h, s)$, following [28]. For a fair comparison, our policy still uses Markovian policy and does not utilize encoded historical messages. PEAC's pseudo-code is in Appendix C.

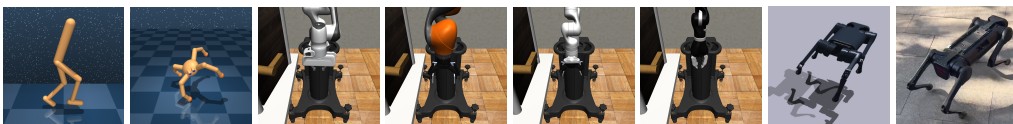

Figure 2: **Benchmark environments,** including DMC [56], Robosuite [73], Isaacgym [37].

## 4 Cross-Embodiment Exploration and Skill Discovery

As shown above, PEAC pre-trains the agent for the optimal initialization to few-shot handle downstream tasks across embodiments. Besides, although PEAC does not directly explore or discover skills, it is flexible to combine with existing unsupervised RL methods, including exploration and skill discovery ones, to achieve cross-embodiment exploration and skill discovery. Below we will discuss in detail the specific combination between PEAC and these two classes respectively, exporting two practical combination algorithms, PEAC-LBS and PEAC-DIAYN, as examples.

**Embodiment-Aware Exploration.** Existing exploration methods mainly encourage the agent to explore unseen regions. As PEAC suggests the agent explores the region where the embodiment discriminator is wrong, it is natural to directly combine $\mathcal{R}_{\text{CE}}$ and exploration intrinsic rewards to achieve cross-embodiment exploration, i.e., balancing embodiment representation learning and unseen state exploration. As an example, we take LBS [38], of which the intrinsic reward is the KL divergence between the latent prior and the approximation posterior, as the PEAC-LBS. As $\mathcal{R}_{\text{CE}}$ and $\mathcal{R}_{\text{LBS}}$ are both related to some KL divergence, we can directly add up these two intrinsic rewards with the same weight in PEAC-LBS, of which the detailed pseudo-code is in Appendix C.

**Embodiment-Aware Skill Discovery.** Single-embodiment skill-discovery mainly maximizes the mutual information between trajectories $\tau$ and skills $z$ as $\mathcal{I}(\tau; z) = D_{\text{KL}}(p(\tau, z) \| p(\tau)p(z))$ [10], which has been shown as optimal initiation to some skill-based adaptation objective [11]. We combine it and our cross-embodiment fine-tuning objective Eq. (2) to propose a unified *cross-embodiment skill-based adaptation objective* as

$$
\begin{aligned}
\mathcal{F}_s(\pi, \pi^*, \mathcal{R}_{\text{ext}}, \boldsymbol{e}) \triangleq{} & \mathbb{E}_{p_{\mathcal{M}_{\boldsymbol{e}}^c, \pi^*}(\tau)}[\mathcal{R}_{\text{ext}}(\tau)] - \max_{\boldsymbol{z}^*} \mathbb{E}_{p_{\mathcal{M}_{\boldsymbol{e}}^c, \pi}(\tau|\boldsymbol{z}^*)}[\mathcal{R}_{\text{ext}}(\tau)] \\
& - \beta D_{\text{KL}}(p_{\mathcal{M}_{\boldsymbol{e}}^c, \pi^*}(\tau) \| p_{\bar{\mathcal{M}}, \pi}(\tau)).
\end{aligned}
\tag{6}
$$

Similar to Theorem 3.2, we can define our pre-training objective and simplify it as

$$
\begin{aligned}
\mathcal{U}_s(\pi, \mathcal{E}) & \triangleq \mathbb{E}_{e \sim \mathcal{E}} \min_{\mathcal{R}_{\text{ext}}(\tau)} \max_{p_{\mathcal{M}_{\boldsymbol{e}}^c, \pi^*}(\tau)} \mathcal{F}_s(\pi, \pi^*, \mathcal{R}_{\text{ext}}, \boldsymbol{e}) \\
& = -\beta \mathbb{E}_{\boldsymbol{e}} \max_{p(\boldsymbol{z}|\mathcal{M}_{\boldsymbol{e}}^c)} \left[ \mathbb{E}_{\tau \sim p_{\mathcal{M}_{\boldsymbol{e}}, \pi}} \log \frac{p_\pi(\boldsymbol{e}|\tau)}{p_\pi(\boldsymbol{e})} + D_{\text{KL}}(p_\pi(\tau, \boldsymbol{z}|\mathcal{M}_{\boldsymbol{e}}^c) \| p_\pi(\boldsymbol{z}|\mathcal{M}_{\boldsymbol{e}}^c) p_\pi(\tau|\mathcal{M}_{\boldsymbol{e}}^c)) \right].
\end{aligned}
\tag{7}
$$

The proof of Eq. (7) is in Appendix A.3, where we also show it is a general form of Theorem 3.2 and the single-embodiment skill-discovery result [11]. The result of Eq. (7) includes two terms for handling cross-embodiment and discovering skills respectively. In detail, the first term is the same as the objective in Eq. (4), thus we can directly optimize it via PEAC. As the second term is similar to the classical skill-discover objective $\mathcal{I}(\tau; z)$ but only embodiment-aware, we can extend existing skill-discovery methods into an embodiment-aware version for handling it.

We take DIAYN [10] as an example, resulting in PEAC-DIAYN. Overall, In the pre-training stage, given a random skill $z$ and an embodiment $e$, we will sample trajectories with the policy $\pi_\theta(\boldsymbol{a}|\boldsymbol{s}, \boldsymbol{z}, \boldsymbol{e})$ that is conditioned on $z$ and the predicted embodiment context. Then we will train a neural network $p(\boldsymbol{z}, \boldsymbol{e}|\tau)$ to jointly predict the current skill and the embodiment. For training the policy, we combine $\mathcal{R}_{\text{CE}}$ and $\mathcal{R}_{\text{DIAYN}}$ as the intrinsic reward. During fine-tuning, we utilize the embodiment discriminator, mapping observed trajectories to infer the embodiment context. We then train an embodiment-aware meta-controller $\pi(\boldsymbol{z}|\boldsymbol{e}, \tau)$, which inputs the state and predicted context and then outputs the skill. It extends existing embodiment-agnostic meta-controller [39] and directly chooses from skill spaces rather than complicated action spaces. The pseudo-code of PEAC-DIAYN is in Appendix C.

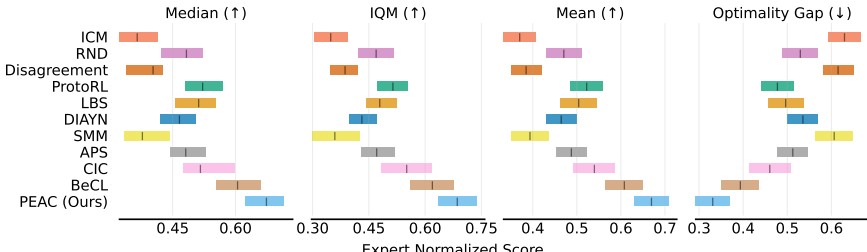

Figure 3: Aggregate metrics [2] in **state-based DMC**. Each statistic for every algorithm has 120 runs (3 embodiment settings × 4 downstream tasks × 10 seeds).

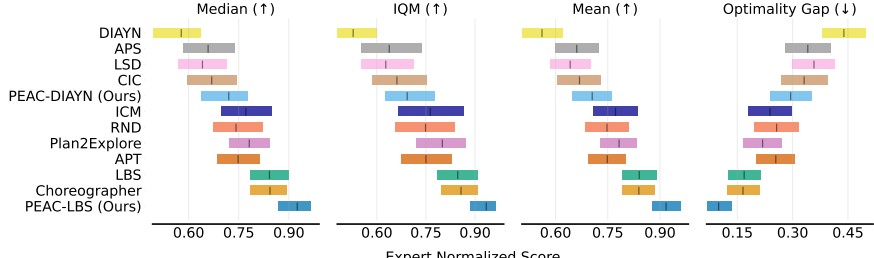

Figure 4: Aggregate metrics [2] in **image-based DMC**. Each statistic for every algorithm has 36 runs (3 embodiment settings × 4 downstream tasks × 3 seeds).

## 5 Experiments

We now present extensive empirical results to answer the following questions:

- Does PEAC enhance the cross-embodiment unsupervised pre-training for handling different downstream tasks? (Sec. 5.2)
- Can CEURL benefit cross-embodiment RL and effectively generalize to unseen embodiments? (Sec. 5.3)
- Does CEURL advantage to real-world cross-embodiment applications? (Sec. 5.4)

### 5.1 Experimental Setup

To fully evaluate PEAC in CEURL, we choose extensive benchmarks (Fig. 2), including state-based / image-based Deepmind Control Suite (DMC) [56] in URLB [27], Robosuite [73, 69] for robotic manipulation, and Isaacgym [37] for simulation as well as real-world legged locomotion. Below we will introduce embodiments, tasks, and baselines for these settings, with more details in Appendix B.

**State-based DMC & Image-based DMC.** These two benchmarks extend URLB [27], classical single-embodiment unsupervised RL settings. Based on basic embodiments, we change the mass or damping to conduct three distinct embodiment distributions: walker-mass, quadruped-mass, and quadruped-damping, following previous work with diverse embodiments [28, 65]. All downstream tasks follow URLB. These two settings take robot states and images as observations respectively.

In state-based DMC, we compare PEAC with 5 exploration and 5 skill-discovery methods: ICM [45], RND [7], Disagreement [46], ProtoRL [62], LBS [38], DIAYN [10], SMM [30], APS [34], CIC [26], and BeCL [61], which are standard and SOTA for this setting. For all methods, we take DDPG [33] as the RL backbone, which is widely used in this benchmark [27]. In image-based DMC, we take 5 exploration baselines: ICM, RND, Plan2Explore [51], APT [35], and LBS; as well as 4 skill-discovery baselines: DIAYN, APS, LSD [42], and CIC. Also, we choose a SOTA baseline Choreographer [39], which combines exploration and skill discovery. For all methods, we take DreamerV2 [18] as the backbone algorithm, which has currently shown leading performance in this benchmark [48].

**Robosuite.** We further consider embodiment distribution with greater change: different robotic arms for manipulation tasks from Robosuite [73]. We pre-train our agents in robotic arms Panda, IIWA, and Kinova3. Besides, we take robotic arm Jaco for evaluating generalization. Following [69], we take DrQ [63] as the RL backbone and choose standard task settings: Door, Lift, and TwoArmPegInHole.

| Domains | Robosuite | | A1-disabled | | | | |
| --- | --- | --- | --- | --- | --- | --- | --- |
| | Train | Test | run | climb | leap | crawl | tilt |
| ICM | 174.4 | 178.3 | 6.7 | 5.7 | 4.0 | 8.5 | 13.9 |
| RND | 171.2 | 185.0 | 8.1 | 3.5 | 2.2 | 7.6 | 6.3 |
| LBS | 157.7 | 166.6 | 0.4 | 1.7 | 1.4 | 1.1 | 2.1 |
| PEAC (Ours) | **190.7** | **200.8** | **19.2** | **10.3** | **10.0** | **20.3** | **17.3** |

Table 1: Results of **Robosuite** and **Isaacgym**.

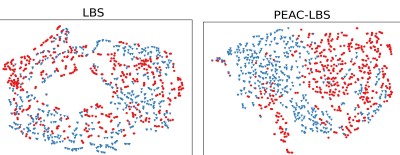

Figure 5: **Generalization Visualization**

| Domains | Walker-mass | Quadruped-mass | Quadruped-damping |
| --- | --- | --- | --- |
| ICM | 391.5 ± 224.9 | 227.1 ± 163.6 | 160.7 ± 129.7 |
| RND | 364.8 ± 172.9 | 588.0 ± 164.8 | 139.5 ± 119.9 |
| Disagreement | 321.6 ± 152.6 | 434.2 ± 176.7 | 140.8 ± 73.9 |
| ProtoRL | 440.1 ± 212.7 | 471.6 ± 209.0 | 328.2 ± 195.4 |
| LBS | 380.3 ± 227.0 | 508.2 ± 222.7 | 350.4 ± 226.2 |
| DIAYN | 267.6 ± 155.3 | 456.6 ± 173.3 | 397.0 ± 159.9 |
| SMM | 451.2 ± 196.6 | 217.3 ± 145.9 | 162.5 ± 119.2 |
| APS | 393.8 ± 222.0 | 464.6 ± 206.0 | 285.5 ± 157.5 |
| CIC | 503.9 ± 260.6 | 602.2 ± 193.8 | 166.2 ± 126.6 |
| BeCL | **544.5 ± 258.7** | 475.6 ± 228.5 | 421.9 ± 246.9 |
| PEAC (Ours) | 491.3 ± 250.1 | **631.0 ± 235.7** | **573.7 ± 220.3** |

| Domains | Walker-mass | Quadruped-mass | Quadruped-damping |
| --- | --- | --- | --- |
| DIAYN | 463.6 ± 250.8 | 399.6 ± 183.7 | 499.9 ± 187.5 |
| APS | 555.5 ± 245.2 | 566.9 ± 158.0 | 546.8 ± 190.6 |
| LSD | 556.6 ± 273.0 | 510.7 ± 173.2 | 520.9 ± 163.8 |
| CIC | 609.4 ± 260.2 | 527.4 ± 229.9 | 558.6 ± 169.5 |
| PEAC-DIAYN (Ours) | 621.9 ± 235.1 | 556.1 ± 179.4 | 557.2 ± 160.7 |
| ICM | 648.1 ± 252.4 | 695.8 ± 180.1 | 590.2 ± 168.9 |
| RND | 658.2 ± 238.8 | 625.7 ± 179.5 | 588.4 ± 175.8 |
| Plan2Explore | 677.4 ± 245.2 | 660.2 ± 162.1 | 608.2 ± 157.7 |
| APT | 643.9 ± 242.6 | 617.7 ± 160.5 | 600.2 ± 149.7 |
| LBS | 658.2 ± 219.7 | 730.7 ± 162.3 | 732.7 ± 142.5 |
| Choreographer | 687.8 ± 222.7 | 682.3 ± 159.4 | 724.6 ± 116.6 |
| PEAC-LBS (Ours) | **754.8 ± 214.6** | **740.8 ± 171.3** | **742.1 ± 165.2** |

Table 2: **Generalization** results of **unseen embodiments** in **state-based DMC** (left) and **image-based DMC** (right). For each domain, we report the average return of each different algorithm and **bold** the best performance.

**Isaacgym.** To explore CEURL in realistic environments, we design embodiment distributions based on the Unitree A1 robot in Isaacgym simulation [37], which is widely used for the real-world legged robot control [1, 74]. As A1 owns 12 controllable joints, we design A1-disabled, a uniform distribution of 12 embodiments, each with a joint failure, respectively. It is realistic as robots may damage some joints when deploying in the real world, and they are still required to complete tasks to their best. We choose standard RL backbone PPO [50] and five downstream tasks: run, climb, leap, crawl, and tilt, following [74]. We take classical baselines for Robosuite and Isaacgym: ICM, RND, and LBS. Besides, we have deployed Aliengo robots with different failure joints to evaluate the effectiveness of PEAC in real-world applications.

## 5.2 Evaluation of PEAC

**State-based DMC.** We first report results in state-based DMC to show that PEAC can facilitate cross-embodiment pre-training. All algorithms, repeated 10 random seeds, are pre-trained 2M timesteps in reward-free environments with different embodiments, followed by fine-tuned downstream tasks for all these embodiments with 100k timesteps. We train DDPG agents for each downstream task 2M steps to get the expert return and calculate the expert normalized score for each method. Following [2], in Fig. 3, we report mean, median, interquartile mean (IQM), and optimality gap (OG) metrics along with stratified bootstrap confidence intervals. Fig. 3 demonstrates that PEAC substantially outperforms other baselines on all metrics, indicating that our cross-embodiment intrinsic reward contributes positively to downstream tasks across different embodiments. Notably, compared with BeCL and CIC which get the second and third scores, PEAC not only has higher performance but also a smaller confidence interval, highlighting its stability. Appendix B.4 reports detailed results of these statistics (Table 8) and individual results for each downstream task (Table 9).

**Image-based DMC.** As described in Sec. 4, PEAC can flexibly combine with existing unsupervised RL methods. To verify it, we evaluate PEAC-LBS and PEAC-DIAYN in image-based DMC. The pre-training and fine-tuning steps are still 2M and 100k respectively. Also, we present four metrics: Median, IQM, Mean, and OG with stratified bootstrap confidence intervals in Fig. 4. Taking IQM as our primary metric, PEAC-LBS not only has a higher value but also a relatively smaller confidence interval, indicating its better stability. As mentioned in [39], pure skill-discovery methods like DIAYN struggle on this benchmark with a certain gap compared to exploratory method. The phenomenon seems more pronounced in cross-embodiment setting than single-embodiment setting, which might be because of the increased difficulty of finding consistent skills across embodiments. As PEAC-DIAYN discovers skills across-embodiment, it consistently leads in performance compared with all other pure skill discovery methods across all four statistics. In Appendix B.5, we report detailed results of these statistics in Table 10 and detailed results for all downstream tasks in Table 11.

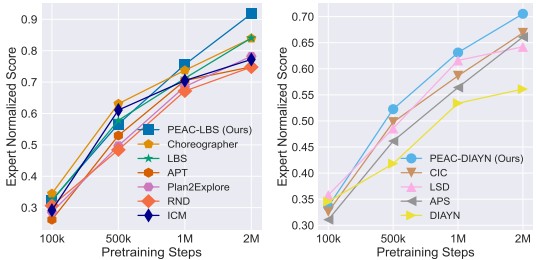
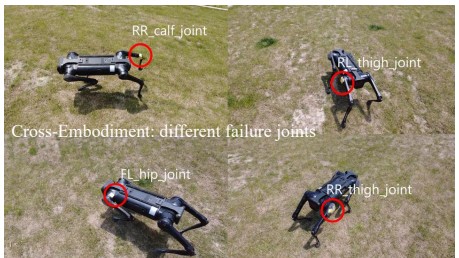

Figure 6: Ablation studies on pre-training timesteps.

Figure 7: Real-world results.

**Robosuite.** Besides, we validate PEAC in a more challenging setting Robosuite where different embodiments own different robotic arms (subfigures 3-6 in Fig. 2). As shown in Table 1, PEAC still significantly outperforms all baselines in both training and testing embodiments, demonstrating its powerful cross-embodiment ability and better generalization ability. The detailed results of each robotic arm are in Table 12 of Appendix B.6.

**Ablation studies.** We do several ablation studies in image-based DMC to clarify the contribution of PEAC better. First, we evaluate the effectiveness of pre-trained steps in fine-tuned performance. We pre-train agents for 100k, 500k, 1M, and 2M steps and then fine-tune them for 100k steps. As shown in Fig. 6, all algorithms improve with pre-training timesteps increasing, indicating that cross-embodiment pre-training effectively benefits fast handling downstream tasks. PEAC-LBS becomes the best-performing method from 1M steps on and PEAC-DIAYN significantly exceeds skill discovery methods. This suggests that PEAC excels at handling cross-embodiment tasks with increased pre-training steps. Additional results are in Appendix B.7. Besides pre-training steps, we also do more ablations studies of different components in PEAC to verify their effectiveness in Appendix B.8. For example, we evaluate the stability of PEAC-LBS in DMC-image under different $\beta$ (we set it as 1.0 in all main experiments), which is the trade-off parameter for balancing the policy improvement term and the policy constraint term. Moreover, we also do an ablation study on our embodiment discriminator to verify the contribution of each component in our PEAC. More results and analyses are in Appendix B.8.

## 5.3 Generalization to Unseen Embodiments

To answer the second question, we further assess the generalization ability of PEAC to unseen embodiments. First, we directly leverage pre-trained agents to zero-shot sample trajectories with different unseen embodiments and then visualize results through t-SNE [57] in Fig. 5, where different colored points represent states sampled via different embodiments. As shown in Fig. 5, PEAC-LBS can distinguish different embodiments' states more effectively compared to LBS, which is difficult to distinguish them (more results are in Appendix B.9). Furthermore, we evaluate the generalization ability of fine-tuned agents for all methods by zero-shot evaluating them with unseen embodiments and the same downstream task. In Table 2, we report the detailed generalization results of all 3 domains about state-based DMC and image-based DMC. The results demonstrate that the fine-tuned agents of PEAC can successfully handle the same downstream task with unseen embodiments, which illustrates that PEAC effectively learns cross-embodiment knowledge. Detailed results for each downstream task are in Appendix B.10 (Table 16-17).

## 5.4 Real-World Applications

To validate CEURL in more realistic settings, we conduct results based on legged locomotion in Isaacgym, which is widely used for real-world applications. First, we present simulation results of A1-disabled in Table 1, with 100M pre-train timesteps and 10M fine-tune timesteps. As shown in Table 1, PEAC effectively establishes a good initialization model across embodiments with different joint failures and quickly adapts to downstream tasks, especially for challenging climb and leap tasks.

Besides, we have deployed PEAC fine-tuned agents in real-world Aliengo-disabled robots, i.e., Aliengo robots with different failure joints. As shown in Fig. 7, due to joint failure, the movement ability of the robot is limited compared to normal settings, but the robot still demonstrates strong adaptability on various terrains not seen in simulators. More images and videos of real-world applications are in Appendix B.12.

## 5.5 Limitations and Discussion

In terms of limitations, we assume that different embodiments may own similar structures so that we can pre-train a unified agent for them. As a result, it might be challenging for PEAC to handle extremely different embodiments. Also, existing unsupervised RL methods still struggle to handle more challenging downstream tasks. In Appendix B.11, we take the first step to evaluate several more challenging downstream tasks and more different embodiment distributions, of which the results show that PEAC can still perform better than baselines. Designing more efficient cross-embodiment unsupervised algorithms for these more difficult and practical settings are interesting future directions. The Broader Impact os this work is discussed in Appendix D.

## 6 Conclusion

In this work, we propose to analyze cross-embodiment RL in an unsupervised RL perspective as CEURL, i.e., pre-training in an embodiment distribution. We formulate it as CE-MDP, with some more challenging properties than the single-embodiment setting. By analyzing the optimal cross-embodiment initialization, we propose PEAC with a principled intrinsic reward function and further show that PEAC can flexibly combine with existing unsupervised RL. Experimental results demonstrate that PEAC can effectively handle downstream tasks across embodiments for extensive settings, ranging from image-based observation, state-based observation, and real-world legged locomotion. We hope this work can encourage further research in developing RL agents for both task generalization and embodiment generalization, especially in real-world control.

## Acknowledgments and Disclosure of Funding

This work was supported by NSFC Projects (Nos. 92248303, 92370124, 62350080, 62276149, 62061136001), BNRist (BNR2022RC01006), Tsinghua Institute for Guo Qiang, and the High Performance Computing Center, Tsinghua University. J. Zhu was also supported by the XPlorer Prize.

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

# A Proof of Theorems

In this section, we will provide detailed proof of theorems in the paper.

## A.1 Properties and Challenges of $\mathcal{D}^{\mathcal{E}}$

*We first construct an example to show that vertices of $\mathcal{D}^{\mathcal{E}}$ may no longer be deterministic policies.*

Considering a simple embodiment distribution with only 2 embodiments $e_1, e_2$ with the embodiment probability $p(e_1) = p(e_2) = \frac{1}{2}$. For each embodiment, there are two states $s_1, s_2$ and two actions $a_1, a_2$ and the dynamic is

$$
\begin{aligned}
&p_{e_1}(s_1|s_1, a_1) = 1, p_{e_1}(s_2|s_1, a_1) = 0, p_{e_1}(s_1|s_1, a_2) = 0, p_{e_1}(s_2|s_1, a_2) = 1 \\
&p_{e_1}(s_1|s_2, a_1) = 0, p_{e_1}(s_2|s_2, a_1) = 1, p_{e_1}(s_1|s_2, a_2) = 1, p_{e_1}(s_2|s_2, a_2) = 0 \\
&p_{e_2}(s_1|s_1, a_1) = 0, p_{e_2}(s_2|s_1, a_1) = 1, p_{e_2}(s_1|s_1, a_2) = 1, p_{e_2}(s_2|s_1, a_2) = 0 \\
&p_{e_2}(s_1|s_2, a_1) = 1, p_{e_2}(s_2|s_2, a_1) = 0, p_{e_2}(s_1|s_2, a_2) = 0, p_{e_2}(s_2|s_2, a_2) = 1,
\end{aligned}
\tag{8}
$$

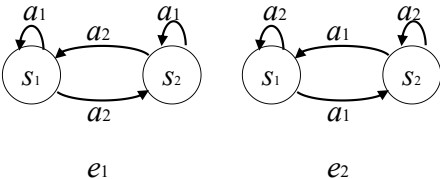

In this setting, there are four deterministic policies:

$$
\begin{aligned}
\pi_1(s_1) = a_1, \pi_1(s_2) = a_1, \quad &\pi_2(s_1) = a_1, \pi_2(s_2) = a_2, \\
\pi_3(s_1) = a_2, \pi_3(s_2) = a_1, \quad &\pi_4(s_1) = a_2, \pi_4(s_2) = a_2.
\end{aligned}
\tag{9}
$$

For any policy $\mu$, we denote that $\rho_{1,\mu}, \rho_{2,\mu}$ are the state distribution of $\mu$ under the environment $\mathcal{P}_{e_1}$ or $\mathcal{P}_{e_2}$ respectively. Then we can calculate that

$$
\begin{aligned}
\rho_{1,\pi_1} &= \left(\frac{1}{2}, \frac{1}{2}\right), \rho_{2,\pi_1} = \left(\frac{1}{2}, \frac{1}{2}\right); \\
\rho_{1,\pi_2} &= \left(\frac{1+\gamma}{2}, \frac{1-\gamma}{2}\right), \rho_{2,\pi_2} = \left(\frac{1-\gamma}{2}, \frac{1+\gamma}{2}\right); \\
\rho_{1,\pi_3} &= \left(\frac{1-\gamma}{2}, \frac{1+\gamma}{2}\right), \rho_{2,\pi_3} = \left(\frac{1+\gamma}{2}, \frac{1-\gamma}{2}\right); \\
\rho_{1,\pi_4} &= \left(\frac{1}{2}, \frac{1}{2}\right), \rho_{2,\pi_4} = \left(\frac{1}{2}, \frac{1}{2}\right).
\end{aligned}
\tag{10}
$$

As the embodiment probability is $p(e_1) = p(e_2) = \frac{1}{2}$, all these four policy share the same state distribution as

$$
\rho_{\pi_1} = \rho_{\pi_2} = \rho_{\pi_3} = \rho_{\pi_4} = \left(\frac{1}{2}, \frac{1}{2}\right).
\tag{11}
$$

Furthermore, we consider a stochastic policy $\pi$ satisfies that

$$
\pi(a_1|s_1) = 1, \pi(a_2|s_1) = 0, \quad \pi(a_1|s_2) = \frac{1}{2}, \pi(a_2|s_2) = \frac{1}{2}.
\tag{12}
$$

Next we will calculate $\rho_{1,\pi}$ and $\rho_{2,\pi}$. For $\rho_{1,\pi}$, at the timestep 0, we have the initial state distribution as $p_0(s_1) = p_0(s_2) = \frac{1}{2}$, assume that at timestep $t$ we have corresponding $p_t(s_1), p_t(s_2)$, we can naturally get the recurrence relation as

$$
p_{t+1}(s_1) = p_t(s_1) + \frac{1}{2}p_t(s_2), \quad p_{t+1}(s_2) = \frac{1}{2}p_t(s_2),
\tag{13}
$$

Naturally, we have $p_t(s_1) = 1 - \frac{1}{2^t}, p_t(s_2) = \frac{1}{2^t}$ and thus the discount state distribution of $s_2$ is

$$
(1-\gamma) \sum_{t=0}^{\infty} \frac{\gamma^t}{2} = \frac{1-\gamma}{2-\gamma}.
\tag{14}
$$

And we have

$$\rho_{1,\pi} = \left( \frac{1}{2-\gamma}, \frac{1-\gamma}{2-\gamma} \right). \tag{15}$$

Similarly, we can calculate $\rho_{2,\pi}$. At the timestep 0, we have the initial state distribution as $p_0(s_1) = p_0(s_2) = \frac{1}{2}$, assume that at timestep $t$ we have corresponding $p_t(s_1), p_t(s_2)$, we can naturally get the recurrence relation as

$$p_{t+1}(s_1) = \frac{1}{2}p_t(s_2), \quad p_{t+1}(s_2) = p_t(s_1) + \frac{1}{2}p_t(s_2), \tag{16}$$

As $p_t(s_1) + p_t(s_2) = 1$, we can solve this recurrence relation via

$$
\begin{aligned}
p_{t+1}(s_1) =& \frac{1}{2}p_t(s_2) = \frac{1}{2} - \frac{1}{2}p_t(s_1), \\
(-2)^{t+1}p_{t+1}(s_1) =& (-2)^t p_t(s_1) - (-2)^t \\
=& ... = (-2)^0 p_0(s_1) - \left( (-2)^t + (-2)^{t-1} + ... + (-2)^0 \right) \\
=& \frac{1}{2} - \frac{1-(-2)^{t+1}}{3} = \frac{1}{6} + \frac{(-2)^{t+1}}{3}, \\
p_{t+1}(s_1) =& \frac{1}{6 \times (-2)^{t+1}} + \frac{1}{3}
\end{aligned}
\tag{17}
$$

Thus the discount state distribution of $s_1$ is

$$(1-\gamma)\sum_{t=0}^{\infty} \gamma^t \left( \frac{1}{6 \times (-2)^t} + \frac{1}{3} \right) = \frac{1}{3} + \frac{1-\gamma}{6}\sum_{t=0}^{\infty}\left( -\frac{\gamma}{2} \right)^t = \frac{1}{3} + \frac{1-\gamma}{6}\frac{1}{1+\frac{\gamma}{2}} = \frac{1}{2+\gamma}. \tag{18}$$

And we have

$$\rho_{2,\pi} = \left( \frac{1}{2+\gamma}, \frac{1+\gamma}{2+\gamma} \right). \tag{19}$$

As the embodiment probability is $p(e_1) = p(e_2) = \frac{1}{2}$, the state distribution of $\pi$ is

$$\rho_\pi = \left( \frac{2}{4-\gamma^2}, \frac{2-\gamma^2}{4-\gamma^2} \right). \tag{20}$$

Taking any $\gamma \in (0,1)$, it is obvious that $\rho_\pi$ is not within the closure composed of $\rho_{\pi_1}, \rho_{\pi_2}, \rho_{\pi_3}, \rho_{\pi_4}$ (actually the point $(1/2,1/2)$). Thus we have explained that the vertices of $\mathcal{D}^{\mathcal{E}}$ might no longer be simple deterministic policies.

### A.2  Proof of Theorem 3.2

*Proof.* Recall that

$$\mathcal{F}(\pi, \pi^*, \mathcal{R}_{\text{ext}}, e) \triangleq \left[ \mathbb{E}_{p_{\mathcal{M}_e^c, \pi^*}(\tau)}[\mathcal{R}_{\text{ext}}(\tau)] - \mathbb{E}_{p_{\mathcal{M}_e^c, \pi}(\tau)}[\mathcal{R}_{\text{ext}}(\tau)] - \beta D_{\text{KL}}(p_{\mathcal{M}_e^c, \pi^*}(\tau) \| p_{\bar{\mathcal{M}}, \pi}(\tau)) \right]. \tag{21}$$

We set a functional $f$ satisfying that

$$f(p(\tau)) = \mathbb{E}_{p(\tau)}[\mathcal{R}_{\text{ext}}(\tau)] - \mathbb{E}_{p_{\mathcal{M}_e^c, \pi}(\tau)}[\mathcal{R}_{\text{ext}}(\tau)] - \beta D_{\text{KL}}(p(\tau) \| p_{\bar{\mathcal{M}}, \pi}(\tau)). \tag{22}$$

Using the calculus of variations, we can calculate its optimal value at the point $p^*$ satisfying that

$$\mathcal{R}_{\text{ext}}(\tau) = \beta \log \frac{p^*(\tau)}{p_{\bar{\mathcal{M}},\pi}(\tau)} + b\beta, \tag{23}$$

here $b$ is a constant not related to $p^*$, and we have $p^*(\tau) = p_{\bar{\mathcal{M}},\pi}(\tau)e^{\frac{\mathcal{R}_{\text{ext}}(\tau)}{\beta} - b}$. As $\int p^*(\tau) = 1$, we can calculate that

$$b = \log \int p_{\bar{\mathcal{M}},\pi}(\tau)e^{\frac{\mathcal{R}_{\text{ext}}(\tau)}{\beta}} d\tau, \quad p^*(\tau) = \frac{p_{\bar{\mathcal{M}},\pi}(\tau)e^{\frac{\mathcal{R}_{\text{ext}}(\tau)}{\beta}}}{\int p_{\bar{\mathcal{M}},\pi}(\tau)e^{\frac{\mathcal{R}_{\text{ext}}(\tau)}{\beta}} d\tau}. \tag{24}$$

Consequently, we have

$$
\max_{p_{\mathcal{M}_e^c,\pi^*}(\tau)} \mathcal{F}(\pi,\pi^*,\mathcal{R}_{\text{ext}},\boldsymbol{e}) = \mathbb{E}_{p^*(\tau)}[\mathcal{R}_{\text{ext}}(\tau)] - \mathbb{E}_{p_{\mathcal{M}_e^c,\pi}(\tau)}[\mathcal{R}_{\text{ext}}(\tau)] - \beta D_{\text{KL}}(p^*(\tau)\|p_{\bar{\mathcal{M}},\pi}(\tau))
$$

$$
= \int p^*(\tau)\mathcal{R}_{\text{ext}}(\tau)d\tau - \mathbb{E}_{p_{\mathcal{M}_e^c,\pi}(\tau)}[\mathcal{R}_{\text{ext}}(\tau)] - \beta \int p^*(\tau)\log\frac{p^*(\tau)}{p_{\bar{\mathcal{M}},\pi}(\tau)}d\tau
$$

$$
= \int p^*(\tau)\mathcal{R}_{\text{ext}}(\tau)d\tau - \mathbb{E}_{p_{\mathcal{M}_e^c,\pi}(\tau)}[\mathcal{R}_{\text{ext}}(\tau)] - \beta \int p^*(\tau)\frac{\mathcal{R}_{\text{ext}}(\tau)}{\beta}d\tau + \beta\log\int p_{\bar{\mathcal{M}},\pi}(\tau)e^{\frac{\mathcal{R}_{\text{ext}}(\tau)}{\beta}}d\tau
$$

$$
= \beta\log\int p_{\bar{\mathcal{M}},\pi}(\tau)e^{\frac{\mathcal{R}_{\text{ext}}(\tau)}{\beta}}d\tau - \mathbb{E}_{p_{\mathcal{M}_e^c,\pi}(\tau)}[\mathcal{R}_{\text{ext}}(\tau)].
$$

$$(25)$$

Similarly, we set a functional $g$ satisfying that

$$
g(r(\tau)) = \beta\log\int p_{\bar{\mathcal{M}},\pi}(\tau)e^{\frac{r(\tau)}{\beta}}d\tau - \mathbb{E}_{p_{\mathcal{M}_e^c,\pi}(\tau)}[r(\tau)]. \tag{26}
$$

Using the calculus of variations, we can calculate its optimal value at the point $r^*$ satisfying that

$$
\beta\frac{\frac{1}{\beta}p_{\bar{\mathcal{M}},\pi}(\tau)e^{\frac{r^*(\tau)}{\beta}}}{\int p_{\bar{\mathcal{M}},\pi}(\tau)e^{\frac{r^*(\tau)}{\beta}}d\tau} = p_{\mathcal{M}_e^c,\pi}(\tau), \quad \frac{r^*(\tau)}{\beta} = \log\frac{p_{\mathcal{M}_e^c,\pi}(\tau)}{p_{\bar{\mathcal{M}},\pi}(\tau)} + \log\int p_{\bar{\mathcal{M}},\pi}(\tau)e^{\frac{r^*(\tau)}{\beta}}d\tau. \tag{27}
$$

Thus we can calculate that

$$
\min_{\mathcal{R}_{\text{ext}}(\tau)}\max_{p_{\mathcal{M}_e^c,\pi^*}(\tau)} \mathcal{F}(\pi,\pi^*,\mathcal{R}_{\text{ext}},\boldsymbol{e}) = \beta\log\int p_{\bar{\mathcal{M}},\pi}(\tau)e^{\frac{r^*(\tau)}{\beta}}d\tau - \mathbb{E}_{p_{\mathcal{M}_e^c,\pi}(\tau)}[r^*(\tau)]
$$

$$
= \beta\log\int p_{\bar{\mathcal{M}},\pi}(\tau)e^{\frac{r^*(\tau)}{\beta}}d\tau - \beta\mathbb{E}_{p_{\mathcal{M}_e^c,\pi}(\tau)}\left[\log\frac{p_{\mathcal{M}_e^c,\pi}(\tau)}{p_{\bar{\mathcal{M}},\pi}(\tau)}\right] - \beta\log\int p_{\bar{\mathcal{M}},\pi}(\tau)e^{\frac{r^*(\tau)}{\beta}}d\tau \tag{28}
$$

$$
= -\beta D_{\text{KL}}\left(p_{\mathcal{M}_e^c,\pi}(\tau)\|p_{\bar{\mathcal{M}},\pi}(\tau)\right),
$$

i.e.,

$$
\mathbb{E}_{\boldsymbol{e}\sim\mathcal{E}}\min_{\mathcal{R}_{\text{ext}}(\tau)}\max_{p_{\mathcal{M}_e,\pi^*}(\tau)} \mathcal{F}(\pi,\pi^*,\mathcal{R}_{\text{ext}},\boldsymbol{e}) = \beta\mathbb{E}_{\boldsymbol{e}\sim\mathcal{E}}\left[-D_{\text{KL}}\left(p_{\mathcal{M}_e^c,\pi}(\tau)\|p_{\bar{\mathcal{M}},\pi}(\tau)\right)\right]
$$

$$
= \beta\mathbb{E}_{\boldsymbol{e}\sim\mathcal{E}}\mathbb{E}_{\tau\sim p_{\mathcal{M}_e^c,\pi}(\tau)}\left[\log\frac{p_\pi(\tau)}{p_\pi(\tau|\mathcal{M}_e^c)}\right] = \beta\mathbb{E}_{\boldsymbol{e}\sim\mathcal{E}}\mathbb{E}_{\tau\sim p_{\mathcal{M}_e^c,\pi}(\tau)}\left[\log\frac{p_\pi(\tau)}{p_\pi(\boldsymbol{e},\tau)/p(\boldsymbol{e})}\right] \tag{29}
$$

$$
= \beta\mathbb{E}_{\boldsymbol{e}\sim\mathcal{E}}\mathbb{E}_{\tau\sim p_{\mathcal{M}_e^c,\pi}(\tau)}\left[\log\frac{p(\boldsymbol{e})}{p_\pi(\boldsymbol{e},\tau)/p_\pi(\tau)}\right] = \beta\mathbb{E}_{\boldsymbol{e}\sim\mathcal{E}}\mathbb{E}_{\tau\sim p_{\mathcal{M}_e^c,\pi}(\tau)}\left[\log\frac{p(\boldsymbol{e})}{p_\pi(\boldsymbol{e}|\tau)}\right].
$$

Thus we have proven this result. $\qquad\square$

### A.3   Detailed Discussion and Proof about Embodiment-Aware Skill Discovery

Here we discuss our *cross-embodiment skill-based adaptation objective*.

**We begin by proving Eq. (7). First, we show that**

$$
\mathbb{E}_{\boldsymbol{e}\sim\mathcal{E}}\min_{\mathcal{R}_{\text{ext}}(\tau)}\max_{p_{\mathcal{M}_e^c,\pi^*}(\tau)} \mathcal{F}_s(\pi,\pi^*,\mathcal{R}_{\text{ext}},\boldsymbol{e})
$$

$$
= -\mathbb{E}_{\boldsymbol{e}}\max_{p(\boldsymbol{z}|\mathcal{M}_e^c)}\mathbb{E}_{\boldsymbol{z}\sim p(\boldsymbol{z}|\mathcal{M}_e^c)}\left[\beta D_{\text{KL}}\left(p_{\mathcal{M}_e^c,\pi}(\tau|\boldsymbol{z})\|p_{\bar{\mathcal{M}},\pi}(\tau)\right)\right]. \tag{30}
$$

Our proof is similar to the proof in Appendix A.2. Recall that

$$
\mathcal{F}_s(\pi,\pi^*,\mathcal{R}_{\text{ext}},\boldsymbol{e}) \triangleq \left[\mathbb{E}_{p_{\mathcal{M}_e^c,\pi^*}(\tau)}[\mathcal{R}_{\text{ext}}(\tau)] - \max_{\boldsymbol{z}^*}\mathbb{E}_{p_{\mathcal{M}_e^c,\pi}(\tau|\boldsymbol{z}^*)}[\mathcal{R}_{\text{ext}}(\tau)] \right.
$$

$$
\left. -\beta D_{\text{KL}}(p_{\mathcal{M}_e^c,\pi^*}(\tau)\|p_{\bar{\mathcal{M}},\pi}(\tau))\right]. \tag{31}
$$

Similar to Eq. (22)-Eq. (25), we have

$$
\max_{p_{\mathcal{M}_e^c,\pi^*}(\tau|\boldsymbol{z})}[\mathbb{E}_{p_{\mathcal{M}_e^c,\pi^*}(\tau|\boldsymbol{z})}[\mathcal{R}_{\text{ext}}(\tau)] - \max_{\boldsymbol{z}^*}\mathbb{E}_{p_{\mathcal{M}_e^c,\pi}(\tau|\boldsymbol{z}^*)}[\mathcal{R}_{\text{ext}}(\tau)] - \beta D_{\text{KL}}(p_{\mathcal{M}_e^c,\pi^*}(\tau|\boldsymbol{z})\|p_{\bar{\mathcal{M}},\pi}(\tau))]
$$

$$
= \beta\log\int p_{\bar{\mathcal{M}},\pi}(\tau)e^{\frac{\mathcal{R}_{\text{ext}}(\tau)}{\beta}}d\tau - \max_{\boldsymbol{z}^*}\mathbb{E}_{p_{\mathcal{M}_e^c,\pi}(\tau|\boldsymbol{z}^*)}[\mathcal{R}_{\text{ext}}(\tau)]
$$

$$
= \min_{\boldsymbol{z}^*}\left[\beta\log\int p_{\bar{\mathcal{M}},\pi}(\tau)e^{\frac{\mathcal{R}_{\text{ext}}(\tau)}{\beta}}d\tau - \mathbb{E}_{p_{\mathcal{M}_e^c,\pi}(\tau|\boldsymbol{z}^*)}[\mathcal{R}_{\text{ext}}(\tau)]\right].
$$

$$(32)$$

Also, similar to Eq. (26)-Eq. (28), we have

$$
\begin{aligned}
&\min_{\mathcal{R}_{\text{ext}}(\tau)} \min_{\boldsymbol{z}^*} \left[ \beta \log \int p_{\bar{\mathcal{M}},\pi}(\tau) e^{\frac{\mathcal{R}_{\text{ext}}(\tau)}{\beta}} d\tau - \mathbb{E}_{p_{\mathcal{M}_{\boldsymbol{e}}^c,\pi}(\tau|\boldsymbol{z}^*)}[\mathcal{R}_{\text{ext}}(\tau)] \right] \\
=&\min_{\boldsymbol{z}^*} \min_{\mathcal{R}_{\text{ext}}(\tau)} \left[ \beta \log \int p_{\bar{\mathcal{M}},\pi}(\tau) e^{\frac{\mathcal{R}_{\text{ext}}(\tau)}{\beta}} d\tau - \mathbb{E}_{p_{\mathcal{M}_{\boldsymbol{e}}^c,\pi}(\tau|\boldsymbol{z}^*)}[\mathcal{R}_{\text{ext}}(\tau)] \right] \\
=&\min_{\boldsymbol{z}^*} \left[ -\beta D_{\text{KL}} \left( p_{\mathcal{M}_{\boldsymbol{e}}^c,\pi}(\tau|\boldsymbol{z}^*) \| p_{\bar{\mathcal{M}},\pi}(\tau) \right) \right].
\end{aligned}
\tag{33}
$$

Thus we have

$$
\begin{aligned}
&\mathbb{E}_{\boldsymbol{e}\sim\mathcal{E}} \min_{\mathcal{R}_{\text{ext}}(\tau)} \max_{p_{\mathcal{M}_{\boldsymbol{e}}^c,\pi^*}(\tau)} \mathcal{F}_s(\pi,\pi^*,\mathcal{R}_{\text{ext}},\boldsymbol{e}) \\
=&\mathbb{E}_{\boldsymbol{e}} \min_{\boldsymbol{z}^*} \left[ -\beta D_{\text{KL}} \left( p_{\mathcal{M}_{\boldsymbol{e}}^c,\pi}(\tau|\boldsymbol{z}^*) \| p_{\bar{\mathcal{M}},\pi}(\tau) \right) \right] \\
=&-\mathbb{E}_{\boldsymbol{e}} \max_{\boldsymbol{z}^*} \left[ \beta D_{\text{KL}} \left( p_{\mathcal{M}_{\boldsymbol{e}}^c,\pi}(\tau|\boldsymbol{z}^*) \| p_{\bar{\mathcal{M}},\pi}(\tau) \right) \right] \\
=&-\mathbb{E}_{\boldsymbol{e}} \max_{p(\boldsymbol{z}|\mathcal{M}_{\boldsymbol{e}}^c)} \mathbb{E}_{\boldsymbol{z}\sim p(\boldsymbol{z}|\mathcal{M}_{\boldsymbol{e}}^c)} \left[ \beta D_{\text{KL}} \left( p_{\mathcal{M}_{\boldsymbol{e}}^c,\pi}(\tau|\boldsymbol{z}) \| p_{\bar{\mathcal{M}},\pi}(\tau) \right) \right],
\end{aligned}
\tag{34}
$$

where the last equality holds from the fact that the maximum is achieved when putting all the probability weight on the input $\boldsymbol{z}$ maximizing $D_{\text{KL}} \left( p_{\mathcal{M}_{\boldsymbol{e}}^c,\pi}(\tau|\boldsymbol{z}) \| p_{\bar{\mathcal{M}},\pi}(\tau) \right)$.

**Next, we will show that** $D_{\text{KL}} \left( p_{\mathcal{M}_{\boldsymbol{e}}^c,\pi}(\tau|\boldsymbol{z}) \| p_{\bar{\mathcal{M}},\pi}(\tau) \right)$ **is a general form of our Theorem 3.2 and the results in the single-embodiment setting [11].** Naturally, when we ignore $\boldsymbol{z}$, $\mathcal{F}_s(\pi,\pi^*,\mathcal{R}_{\text{ext}},\boldsymbol{e})$ will degenerate into $\mathcal{F}(\pi,\pi^*,\mathcal{R}_{\text{ext}},\boldsymbol{e})$, and Eq. (7) will also degenerate into Eq. (4), i.e., the results in Theorem 3.2. On the other hand, if we change Eq. (7) into the single-embodiment setting, i.e., $\mathcal{E}$ is a Dirac distribution with the probability $p(\boldsymbol{e})=1$ for some fixed $\boldsymbol{e}$, then we have

$$
\begin{aligned}
&\max_{\pi} \min_{\mathcal{R}_{\text{ext}}(\tau)} \max_{p_{\mathcal{M}_{\boldsymbol{e}}^c,\pi^*}(\tau)} \mathcal{F}_s(\pi,\pi^*,\mathcal{R}_{\text{ext}},\boldsymbol{e}) \\
=&\max_{\pi} \left[ -\max_{p(\boldsymbol{z}|\mathcal{M}_{\boldsymbol{e}}^c)} \mathbb{E}_{\boldsymbol{z}\sim p(\boldsymbol{z}|\mathcal{M}_{\boldsymbol{e}}^c)} \left[ \beta D_{\text{KL}} \left( p_{\mathcal{M}_{\boldsymbol{e}}^c,\pi}(\tau|\boldsymbol{z}) \| p_{\mathcal{M}_{\boldsymbol{e}}^c,\pi}(\tau) \right) \right] \right] \\
=&-\min_{\pi} \max_{p(\boldsymbol{z})} \mathbb{E}_{\boldsymbol{z}\sim p(\boldsymbol{z})} \left[ \beta D_{\text{KL}} \left( p_{\mathcal{M}_{\boldsymbol{e}}^c,\pi}(\tau|\boldsymbol{z}) \| p_{\mathcal{M}_{\boldsymbol{e}}^c,\pi}(\tau) \right) \right] \\
\approx&-\min_{\rho} \max_{p(\boldsymbol{z})} \mathbb{E}_{\boldsymbol{z}\sim p(\boldsymbol{z})} \left[ \beta D_{\text{KL}} \left( p(\tau|\boldsymbol{z}) \| \rho(\tau) \right) \right],
\end{aligned}
\tag{35}
$$

the last approximation simplifies the complex coupling relationship between $\pi$ and $\boldsymbol{z}$, following [11]. Furthermore, by Lemma 6.5 in [11] (proof in Theorem 13.1.1 from [9]), we have

$$
\min_{\rho} \max_{p(\boldsymbol{z})} \mathbb{E}_{\boldsymbol{z}\sim p(\boldsymbol{z})} \left[ D_{\text{KL}} \left( p(\tau|\boldsymbol{z}) \| \rho(\tau) \right) \right] = \max_{p(\boldsymbol{z})} \mathcal{I}(\tau;\boldsymbol{z}),
\tag{36}
$$

which is the objective of existing single-embodiment skill-discovery methods.

**Finally, we will Eq. (7), which further indicates that our cross-embodiment skill-based objective can be decomposed into two terms: one for handling cross-embodiment while the other aims at discovering skills.** Actually, we have

$$
\begin{aligned}
&\mathbb{E}_{\boldsymbol{e}\sim\mathcal{E}} \min_{\mathcal{R}_{\text{ext}}(\tau)} \max_{p_{\mathcal{M}_{\boldsymbol{e}}^c,\pi^*}(\tau)} \mathcal{F}_s(\pi,\pi^*,\mathcal{R}_{\text{ext}},\boldsymbol{e}) \\
=&\mathbb{E}_{\boldsymbol{z}\sim p(\boldsymbol{z}|\mathcal{M}_{\boldsymbol{e}}^c)} \left[ D_{\text{KL}} \left( p_{\mathcal{M}_{\boldsymbol{e}}^c,\pi}(\tau|\boldsymbol{z}) \| p_{\bar{\mathcal{M}},\pi}(\tau) \right) \right] \\
=&\int \frac{p_\pi(\boldsymbol{e},\tau,\boldsymbol{z})}{p(\boldsymbol{e})} \log \frac{p_\pi(\tau|\boldsymbol{z},\boldsymbol{e})}{p_\pi(\tau)} d\boldsymbol{z} d\tau = \int p_\pi(\tau,\boldsymbol{z}|\boldsymbol{e}) \log \frac{p_\pi(\boldsymbol{z},\boldsymbol{e}|\tau)}{p_\pi(\boldsymbol{e},\boldsymbol{z})} d\boldsymbol{z} d\tau \\
=&\int p_\pi(\tau,\boldsymbol{z}|\boldsymbol{e}) \log \frac{p_\pi(\boldsymbol{e}|\tau)p_\pi(\boldsymbol{z}|\boldsymbol{e},\tau)}{p_\pi(\boldsymbol{e})p_\pi(\boldsymbol{z}|\boldsymbol{e})} d\boldsymbol{z} d\tau \\
=&\int p_\pi(\tau|\boldsymbol{e}) \log \frac{p_\pi(\boldsymbol{e}|\tau)}{p_\pi(\boldsymbol{e})} d\tau + \int p_\pi(\tau,\boldsymbol{z}|\boldsymbol{e}) \log \frac{p_\pi(\tau,\boldsymbol{z}|\boldsymbol{e})}{p_\pi(\boldsymbol{z}|\boldsymbol{e})p_\pi(\tau|\boldsymbol{e})} d\boldsymbol{z} d\tau \\
=&\mathbb{E}_{\tau\sim p_{\mathcal{M}_{\boldsymbol{e}}^c,\pi}} \left[ \log \frac{p_\pi(\boldsymbol{e}|\tau)}{p_\pi(\boldsymbol{e})} + D_{\text{KL}}(p_\pi(\tau,\boldsymbol{z}|\boldsymbol{e}) \| p_\pi(\boldsymbol{z}|\boldsymbol{e})p_\pi(\tau|\boldsymbol{e})) \right].
\end{aligned}
\tag{37}
$$

# B Experimental Details

In this section, we will introduce more detailed information about our experiments. In Sec. B.1, we introduce the detailed environments and tasks used in our experiments. In Sec. B.2, we will illustrate all the baselines compared in experiments. Also, all hyper-parameters of experiments are in Sec. B.3. Moreover, we supplement more detailed experimental results about state-based DMC, image-based DMC, and Robosuite in Sec. B.4, Sec. B.5, and Sec. B.6, respectively. Then we conduct detailed generalization results of pre-trained models and fine-tuned models in Sec. B.9 and Sec. B.10, respectively. Finally, we report more detailed real-world experiments in Sec. B.12.

## B.1 Embodiments and Tasks

**State-based DMC.** This benchmark is based on DMC [56] and URLB [27] with state-based observation. Each domain contains one robot and four downstream tasks. We extend it into the cross-embodiment settings: Walker-mass, Quadruped-mass, and Quadruped-damping. Walker-mass extends the Walker robot in DMC, which is a two-leg robot, and designs a distribution with different mass $m$, i.e., $m$ times the mass of a standard walker robot. Similarly, Quadruped-mass also considers quadruped robots with different mass $m$. Quadruped-damping, on the other hand, changes the damping of the standard quadruped robot with $l$ times. The detailed parameters of training embodiments and generalization embodiments are in Table 3.

**Image-based DMC.** This benchmark is the same with state-based DMC but with image-based observation. Thus we consider similar three embodiment distributions: Walker-mass, Quadruped-mass, and Quadruped-damping.

|  | Train | Generalization |
|---|---|---|
| Walker-mass | $m \in \{0.2, 0.6, 1.0, 1.4, 1.8\}$ | $m \in \{0.4, 0.8, 1.2, 1.6\}$ |
| Quadruped-mass | $m \in \{0.4, 0.8, 1.0, 1.4\}$ | $m \in \{0.6, 1.2\}$ |
| Quadruped-damping | $l \in \{0.2, 0.6, 1.0, 1.4, 1.8\}$ | $l \in \{0.4, 0.8, 1.2, 1.6\}$ |

Table 3: Environment parameters used for state-based DMC and image-based DMC.

**Robosuite.** This benchmark utilizes the environment in [73] and follows the experimental setting in RL-Vigen [69], of which the cross-embodiment setting includes Panda, IIWA, and Kinova3. Here different embodiments may own different shapes (observations), and dynamics. Similarly, we pre-train cross-embodiments in all these three embodiments and fast fine-tune the pre-trained agents to downstream tasks. Besides these three embodiments, we also directly fine-tune our pre-trained models in one unseen embodiment: Jaco, to validate the cross-embodiment generalization ability of CEURL. For task sets, we consider three widely used tasks: Door, Lift, and TwoArmPegInHole. Noticing that although these three tasks can be finished by the same robots, their demand for robotic arms varies a lot. For example, TwoArmPegInHole needs two robotic arms but the other two tasks only need one. Consequently, we pre-train cross-embodiment agents for each single task, for all methods.

**Isaacgym.** We first design a setting in simulation based on Unitree A1 in Isaacgym, which is a challenging legged locomotion task and is widely used for real-world legged locomotion. The action space of A1 is a 12-dimension vector, representing 12 joint torque. Thus we consider our A1-disabled benchmark, including 12 embodiments, each of which owns a joint torque failure, i.e., the torque output of this joint is always 0 in this embodiment. This setting is practical as our robot may experience partial joint failure during use, and we still hope that it can complete the task as much as possible.

Moreover, we deploy PEAC into real-world Aliengo robots with failure joints. Similarly, we consider the embodiment distribution Aliengo-disabled, which owns 12 embodiments, each of which owns a joint torque failure respectively. We first pre-train a unified agent across these 12 embodiment in reward-free environments. During fine-tuning, for each embodiment, we utilize the same pre-trained agent to fine-tune the given moving task through this embodiment. Finally, we deploy the fine-tuned agent into the real-world setting to evaluate its movement ability under different kinds of terrains with joint failure.

## B.2 Baselines and Implementations

**ICM [45].** Intrinsic Curiosity Module (ICM) designs intrinsic rewards as the divergence between the projected state representations in a feature space and the estimations made by a feature dynamics model.

**RND [7].** Random Network Distillation (RND) utilizes a predictor network's error in imitating a randomly initialized target network to generate intrinsic rewards, enhancing exploration in learning environments.

**Disagreement [46] / Plan2Explore [51].** The Disagreement algorithm leverages prediction variance across multiple models to estimate state uncertainty, guiding exploration towards less certain states. The Plan2Explore algorithm employs a self-supervised, world-model-based framework, using model disagreement to assess environmental uncertainty and incentivize exploration in sparse-reward scenarios.

**ProtoRL [62].** Proto-RL combines representation learning and exploration through a self-supervised learning framework, using prototype representations to pre-train task-independent representations in the environment, effectively improving policy learning in continuous control tasks.

**APT [35].** Active Pre-training (APT) estimates entropy for a given state using a particle-based estimator based on the K nearest-neighbors algorithm.

**LBS [38].** Latent Bayesian Surprise (LBS) applies Bayesian surprise within a latent space, efficiently facilitating exploration by measuring the disparity between an agent's prior and posterior beliefs about system dynamics.

**Choreographer [39].** Choreographer is a model-based approach in unsupervised skill learning that employs a world model for skill acquisition and adaptation, distinguishing exploration from skill learning and leveraging a meta-controller for efficient skill adaptation in simulated scenarios, enhancing adaptability to downstream tasks and environmental exploration.

**DIAYN [10].** Diversity is All You Need (DIAYN) autonomously learns a diverse set of skills by maximizing mutual information between states and latent skills, using a maximum entropy policy.

**SMM [30].** State Marginal Matching (SMM) develops a task-agnostic exploration strategy by learning a policy to match the state distribution of an agent with a given target state distribution.

**APS [34].** Active Pre-training with Successor Feature (APS) maximizes the mutual information between states and task variables by reinterpreting and combining variational successor features with nonparametric entropy maximization.

**LSD [42].** Lipschitz-constrained Skill Discovery (LSD) adopts a Lipschitz-constrained state representation function, ensuring that maximizing this objective in the latent space leads to an increase in traveled distances or variations in the state space, thereby enabling the discovery of more diverse, dynamic, and far-reaching skills.

**CIC [26].** Contrastive Intrinsic Control (CIC) is an unsupervised reinforcement learning algorithm that leverages contrastive learning to maximize the mutual information between state transitions and latent skill vectors, subsequently maximizing the entropy of these embeddings as intrinsic rewards to foster behavioral diversity.

**BeCL [61].** Behavior Contrastive Learning (BeCL) utilizes contrastive learning for unsupervised skill discovery, defining its reward function based on the mutual information between states generated by the same skill.

Next, we will introduce the implementations of baselines for all experimental settings.

For **state-based DMC**, almost all baselines (ICM, RND, Disagreement, ProtoRL, DIAYN, SMM, APS) combined with RL backbone DDPG are directly following the official implementation in urlb (`https://github.com/rll-research/url_benchmark`). For LBS, we refer the implementation in [48] (`https://github.com/mazpie/mastering-urlb`) and combine it with the code of urlb. For other more recent baselines, we also follow their official implementations, including CIC (`https://github.com/rll-research/cic`) and BeCL (`https://github.com/Rooshy-yang/BeCL`).

For **image-based DMC**, almost all baselines (ICM, RND, Plan2Explore, APT, LBS, DIAYN, APS) combined with RL backbone DreamerV2 are directly following the official implementation in [48] (`https://github.com/mazpie/mastering-urlb`), which currently achieves the leading performance in image-based DMC of urlb. For CIC, we combine its official code (`https://github.com/rll-research/cic`), which mainly considers state-based DMC, and the DreamerV2 backbone in [48]. Similarly, for LSD, we refer to its official code (`https://github.com/seohongpark/LSD`) and combine it with the code of [48]. For Choreographer, of which the backbone is DreamerV2, we directly utilize its official code (`https://github.com/mazpie/choreographer`).

For **Robosuite**, our code is based on the code of RL-Vigen [69] (`https://gemcollector.github.io/RL-ViGen`), including the RL backbone DrQ. For **Isaacgym**, our code is based on the official code of [74] (`https://github.com/ZiwenZhuang/parkour`), which implements five downstream tasks (run, climb, leap, crawl, tilt). For these two settings (Robosuite and Isaacgym), as there are few works considering unsupervised RL in such a challenging setting, we implement classical baselines (ICM, RND, LBS) by referring their implementations in urlb (`https://github.com/rll-research/url_benchmark`) and [48] (`https://github.com/mazpie/mastering-urlb`).

## B.3   Hyper-parameters

Baseline hyper-parameters are taken from their implementations (see Appendix B.2 above). Here we introduce PEAC's hyper-parameters. For all settings, hyper-parameters of RL backbones (DDPG, DreamerV2, PPO) follow standard settings.

First, for PEAC in state-based DMC with RL backbone DDPG, our code is based on urlb (`https://github.com/mazpie/mastering-urlb`) and inherits hyper-parameters of DDPG. For completeness, we list all hyper-parameters as

| DDPG Hyper-parameter | Value |
|---|---|
| Replay buffer capacity | $10^6$ |
| Action repeat | 1 |
| Seed frames | 4000 |
| n-step returns | 3 |
| Mini-batch size | 1024 |
| Seed frames | 4000 |
| Discount $\gamma$ | 0.99 |
| Optimizer | Adam |
| Learning rate | 1e-4 |
| Agent update frequency | 2 |
| Critic target EMA rate $\tau_Q$ | 0.01 |
| Features dim. | 1024 |
| Hidden dim. | 1024 |
| Exploration stddev clip | 0.3 |
| Exploration stddev value | 0.2 |
| Number pre-training frames | $2 \times 10^6$ |
| Number fine-turning frames | $1 \times 10^5$ |
| **PEAC Hyper-parameter** | **Value** |
| Historical information encoder | GRU $(\dim(\mathcal{S}) + \dim(\mathcal{A}) \to 1024)$ |
| Encoded historical information length | 10 |
| Embodiment context model | MLP $(1024 \to \text{Embodiment context dim})$ |

Table 4: Details of hyper-parameters used for state-based DMC.

Next, for PEAC-LBS and PEAC-DIAYN in image-based DMC with RL backbone DreamerV2, our code is based on [48] (`https://github.com/mazpie/mastering-urlb`). Hyper-parameters of

PEAC-LBS and PEAC-DIAYN inherit DreamerV2's hyper-parameters, as well as inherit hyper-parameters of LBS and DIAYN, respectively.

| DreamerV2 Hyper-parameter | Value |
|---|---|
| Environment frames/update | 10 |
| MLP number of layers | 4 |
| MLP number of units | 400 |
| Hidden layers dimension | 400 |
| Adam epsilon | $1 \times 10^{-5}$ |
| Weight decay | $1 \times 10^{-6}$ |
| Gradient clipping | 100 |
| World Model | |
| Batch size | 50 |
| Sequence length | 50 |
| Discrete latent state dimension | 32 |
| Discrete latent classes | 32 |
| GRU cell dimension | 200 |
| KL free nats | 1 |
| KL balancing | 0.8 |
| Adam learning rate | $3 \times 10^{-4}$ |
| Slow critic update interval | 100 |
| Actor-Critic | |
| Imagination horizon | 15 |
| Discount $\gamma$ | 0.99 |
| GAE $\lambda$ | 0.95 |
| Adam learning rate | $8 \times 10^{-5}$ |
| Actor entropy loss scale | $1 \times 10^{-4}$ |
| **PEAC-LBS Hyper-parameter** | **Value** |
| Embodiment context model | MLP (DreamerV2 encoder dim $\rightarrow 200 \rightarrow 200 \rightarrow$ Embodiment context dim) |
| LBS model | MLP (DreamerV2 encoder dim $\rightarrow 200 \rightarrow 200 \rightarrow 200 \rightarrow 200 \rightarrow 1$) |
| **PEAC-DIAYN Hyper-parameter** | **Value** |
| Embodiment context model | MLP (DreamerV2 encoder dim $\rightarrow 200 \rightarrow 200 \rightarrow$ Embodiment context dim) |
| DIAYN model | MLP (DreamerV2 encoder dim $\rightarrow 200 \rightarrow 200 \rightarrow$ skill dim) |

Table 5: Details of hyper-parameters used for image-based DMC.

Then, for PEAC in Robosuite, our code follows RL-Vigen [69] (`https://gemcollector.github.io/RL-ViGen`). PEAC's hyper-parameters, inheriting DrQ's hyperparameters, include

| DrQ Hyper-parameter | Value |
|---|---|
| Discount factor | 0.99 |
| Optimizer | Adam |
| Learning rate | 1e-4 |
| Action repeat | 1 |
| N-step return | 1 |
| Hidden dim | 1024 |
| Frame stack | 3 |
| Replay Buffer size | 1000000 |
| Feature dim | 50 |
| **PEAC Hyper-parameter** | **Value** |
| Historical information encoder | GRU (Encoder Feature Dim + $\dim(\mathcal{A}) \rightarrow 50$) |
| Encoded historical information length | 10 |
| Embodiment context model | MLP ($50 \rightarrow$ Embodiment context dim) |

Table 6: Details of hyper-parameters used for Robosuite.

Finally, for PEAC in A1-disabled of Isaacgym with RL backbone PPO, our code follows [74] (`https://github.com/ZiwenZhuang/parkour`). PEAC's hyper-parameters, inheriting PPO's hyperparameters, include

| PPO Hyper-parameter | Value |
|---|---|
| PPO clip range | 0.2 |
| GAE $\lambda$ | 0.95 |
| Learning rate | 1e-4 |
| Reward discount factor | 0.99 |
| Minimum policy std | 0.2 |
| Number of environments | 4096 |
| Number of environment steps per training batch | 24 |
| Learning epochs per training batch | 5 |
| Number of mini-batches per training batch | 4 |
| **PEAC Hyper-parameter** | **Value** |
| Historical information encoder | GRU ($\dim(\mathcal{S}) + \dim(\mathcal{A}) \to 128$) |
| Encoded historical information length | 24 |
| Embodiment context model | MLP ($128 \to$ Embodiment context dim) |

Table 7: Details of hyper-parameters used for Isaacgym.

## B.4 Detailed results in state-based DMC

In Table 8, we present detailed results in state-based DMC of all four statistics (medium, IQM, mean, OG) for baselines and our PEAC. The results indicate that PEAC performs the best in all these four metrics, while BeCL and CIC perform second and third respectively. Moreover, we report individual results for each downstream task of state-based DMC in Table 9. PEAC performs comparably to BeCL as well as CIC in the Walker-mass tasks and best on most Quadruped-mass and Quadruped-damping tasks. Especially, in the challenging Quadruped-damping setting, PEAC can complete cross-embodiment downstream tasks and significantly outperforms BeCL and CIC.

| Metrics | Median | IQM | Mean | Optimality Gap |
|---|---|---|---|---|
| ICM | 0.37 | 0.35 | 0.37 | 0.63 |
| RND | 0.48 | 0.47 | 0.47 | 0.53 |
| Disagreement | 0.40 | 0.39 | 0.38 | 0.62 |
| ProtoRL | 0.52 | 0.51 | 0.52 | 0.48 |
| LBS | 0.51 | 0.48 | 0.50 | 0.50 |
| DIAYN | 0.47 | 0.43 | 0.46 | 0.54 |
| SMM | 0.38 | 0.36 | 0.39 | 0.61 |
| APS | 0.48 | 0.47 | 0.49 | 0.51 |
| CIC | 0.52 | 0.55 | 0.54 | 0.46 |
| BeCL | 0.60 | 0.62 | 0.61 | 0.39 |
| PEAC (Ours) | **0.67** | **0.69** | **0.67** | **0.33** |

Table 8: **Aggregate metrics [2] in state-based DMC**. For every algorithm, there are 3 embodiment settings, each trained with 10 seeds and fine-tuned under 4 downstream tasks, thus each statistic for every method has 120 runs.

| Domains | Walker-mass | | | | Quadruped-mass | | | | Quadruped-damping | | | | Normilized |
|---|---|---|---|---|---|---|---|---|---|---|---|---|---|
| Tasks | stand | walk | run | flip | stand | walk | run | jump | stand | walk | run | jump | Average |
| ICM | 665.3 | 418.0 | 146.2 | 246.6 | 460.2 | 229.5 | 215.6 | 323.5 | 365.8 | 182.4 | 180.2 | 203.1 | 0.37 |
| RND | 588.9 | 386.7 | 176.4 | 253.8 | **820.6** | 563.7 | 409.6 | 589.5 | 325.4 | 166.2 | 156.0 | 235.8 | 0.47 |
| Disagreement | 549.3 | 331.6 | 139.8 | 250.0 | 555.5 | 372.4 | 329.8 | 506.1 | 274.0 | 139.1 | 142.6 | 217.2 | 0.38 |
| ProtoRL | 731.6 | 458.0 | 192.0 | 325.8 | 687.0 | 430.0 | 348.7 | 514.3 | 498.3 | 336.4 | 275.2 | 364.1 | 0.52 |
| LBS | 618.0 | 370.3 | 136.8 | 343.1 | 740.8 | 499.1 | 388.7 | 517.2 | 574.0 | 302.0 | 258.4 | 335.8 | 0.50 |
| DIAYN | 502.1 | 245.2 | 106.8 | 212.7 | 682.7 | 484.3 | 371.0 | 469.1 | 553.4 | 386.7 | 331.8 | 394.8 | 0.46 |
| SMM | 673.5 | 509.2 | 220.7 | 329.6 | 357.0 | 176.4 | 189.7 | 277.8 | 314.2 | 174.0 | 183.0 | 287.5 | 0.39 |
| APS | 629.8 | 429.8 | 129.4 | 291.4 | 653.1 | 474.1 | 325.3 | 533.7 | 479.9 | 254.9 | 302.4 | 403.7 | 0.49 |
| CIC | 824.8 | 536.6 | 220.7 | 327.7 | 762.5 | 610.9 | **442.7** | 617.5 | 335.9 | 194.1 | 166.4 | 267.5 | 0.54 |
| BeCL | **838.6** | **623.6** | **238.5** | **348.1** | 729.8 | 445.0 | 349.4 | 557.1 | 553.7 | 485.8 | 292.0 | 509.8 | 0.61 |
| PEAC (Ours) | 823.8 | 499.9 | 210.6 | 320.5 | 786.0 | 754.5 | 388.3 | 645.6 | 712.3 | 644.1 | 393.5 | 541.8 | **0.67** |

Table 9: **Detailed results in state-based DMC**. Average cumulative reward (mean of 10 seeds) of the best policy.

## B.5 Detailed results in image-based DMC

In Table 10, we present detailed results in state-based DMC of all four statistics (medium, IQM, mean, OG) for baselines and our PEAC-LBS as well as PEAC-DIAYN. Besides these statistics, in Table 11, we further report the detailed results for the 12 downstream tasks, averaged across all embodiments and seeds. Overall, PEAC-LBS's performance is steadily on top, outperforming existing methods, especially in Walker-mass. Also, compared with other pure skill discovery methods, PEAC-DIAYN performs more consistently on all tasks and achieves higher average rewards.

| Metrics | Median | IQM | Mean | Optimality Gap |
|---|---|---|---|---|
| DIAYN | 0.58 | 0.53 | 0.56 | 0.44 |
| APS | 0.66 | 0.64 | 0.66 | 0.34 |
| LSD | 0.64 | 0.63 | 0.64 | 0.36 |
| CIC | 0.67 | 0.66 | 0.67 | 0.33 |
| PEAC-DIAYN (Ours) | 0.72 | 0.69 | 0.71 | 0.29 |
| ICM | 0.77 | 0.76 | 0.77 | 0.24 |
| RND | 0.74 | 0.75 | 0.75 | 0.26 |
| Plan2Explore | 0.78 | 0.80 | 0.78 | 0.22 |
| APT | 0.75 | 0.75 | 0.75 | 0.25 |
| LBS | 0.84 | 0.85 | 0.84 | 0.17 |
| Choreographer | 0.84 | 0.86 | 0.84 | 0.17 |
| PEAC-LBS (Ours) | **0.93** | **0.93** | **0.92** | **0.10** |

Table 10: **Aggregate metrics [2] in image-based DMC**. For every algorithm, there are 3 embodiment settings, each trained with 3 seeds and fine-tuned under 4 downstream tasks, thus each statistic for every method has 36 runs.

| Domains | Walker-mass | | | | Quadruped-mass | | | | Quadruped-damping | | | | Normalized |
|---|---|---|---|---|---|---|---|---|---|---|---|---|---|
| Tasks | stand | walk | run | flip | stand | walk | run | jump | stand | walk | run | jump | Average |
| DIAYN | 772.6 | 515.1 | 193.8 | 365.7 | 583.9 | 425.9 | 311.9 | 431.8 | 791.8 | 410.8 | 367.4 | 536.1 | 0.56 |
| APS | 906.2 | 554.1 | 228.9 | 473.2 | 814.9 | 414.8 | 413.5 | 677.2 | 850.5 | 417.0 | 379.1 | 560.7 | 0.66 |
| LSD | 912.8 | 644.1 | 227.9 | 401.9 | 769.0 | 409.2 | 401.3 | 555.5 | 634.9 | 447.9 | 481.4 | 608.5 | 0.64 |
| CIC | 930.5 | 725.7 | 289.8 | 423.6 | 850.3 | 410.4 | 341.8 | 488.2 | 883.3 | 457.0 | 416.5 | 572.3 | 0.67 |
| PEAC-DIAYN (Ours) | 954.5 | 731.8 | 305.9 | 491.1 | 720.5 | 420.1 | 446.6 | 548.8 | 867.3 | 503.6 | 440.8 | 671.6 | 0.71 |
| ICM | 946.5 | 797.0 | 304.6 | 493.8 | **937.4** | 610.4 | 461.0 | 809.7 | 834.9 | 458.0 | 438.7 | 683.3 | 0.77 |
| RND | 950.1 | 749.2 | 326.7 | 510.6 | 903.9 | 509.5 | 444.4 | 733.5 | 814.3 | 444.5 | 405.5 | 708.6 | 0.75 |
| Plan2Explore | 956.5 | 836.0 | 342.2 | 518.8 | 895.7 | 652.4 | 470.9 | 634.5 | 890.9 | 583.8 | 421.2 | 689.7 | 0.78 |
| APT | 914.2 | 781.1 | 332.8 | 485.5 | 833.6 | 513.4 | 489.2 | 718.3 | 863.8 | 494.4 | 450.1 | 639.1 | 0.75 |
| LBS | 937.9 | 754.4 | 365.1 | 531.2 | 900.1 | 732.3 | 535.1 | 777.4 | 883.4 | 731.7 | 511.1 | 758.5 | 0.84 |
| Choreographer | 957.8 | 819.4 | 368.3 | 551.6 | 913.2 | 686.1 | 459.8 | 757.1 | 888.0 | 715.6 | 590.1 | 706.8 | 0.84 |
| PEAC-LBS (Ours) | **964.5** | **892.1** | **418.3** | **673.5** | 917.7 | **744.5** | **607.7** | **814.9** | **908.3** | **775.9** | **648.2** | **784.1** | **0.92** |

Table 11: **Detailed results in image-based DMC**. Average cumulative reward (mean of 3 seeds) of the best policy trained by different algorithms. We **bold** the best performance of each task. The six baselines above are exploration-based methods (Choreographer utilizes both exploration and skill-discovery techniques), while the following four baselines are skill-discovery methods.

## B.6 Detailed results in Robosuite

In Table 12, we report detailed results in Robosuite with all tasks and robotic arms. Overall, PEAC performs better in more tasks and owns better generalization ability to unseen robot Jaco.

| Domains | Door | | | | Lift | | | | TwoArmPegInHole | | | |
|---|---|---|---|---|---|---|---|---|---|---|---|---|
| | Panda | IIWA | Kinova3 | Jaco | Panda | IIWA | Kinova3 | Jaco r | Panda | IIWA | Kinova3 | Jaco |
| ICM | 156.2 | 134.4 | 32.2 | 107.7 | **134.1** | 151.6 | 85.9 | 89.5 | 288.4 | **282.8** | 304.1 | 337.8 |
| RND | 128.4 | 150.5 | **148.0** | 127.2 | 74.0 | 92.7 | 84.4 | 64.2 | 272.7 | 277.6 | **312.8** | **363.5** |
| LBS | 120.4 | 128.7 | 79.6 | 104.0 | 89.7 | 80.2 | 66.7 | 87.9 | 268.0 | 271.5 | **314.9** | 308.0 |
| PEAC (Ours) | **225.4** | **158.1** | 112.4 | **161.9** | 109.8 | 140.1 | **92.2** | 118.5 | **285.5** | 281.3 | 311.4 | 321.9 |

Table 12: **Detailed results in Robosuite**.

### B.7 Ablation of timesteps in image-based DMC

In Figure 8, we show additional results about the performance in three domains of image-based DMC for different algorithms and pre-training timesteps. Overall, PEAC-LBS outperforms all methods, while Choreographer and LBS are still competitive on the Quadruped-mass. Also, PEAC-DIAYN outperforms all other pure skill discovery methods.

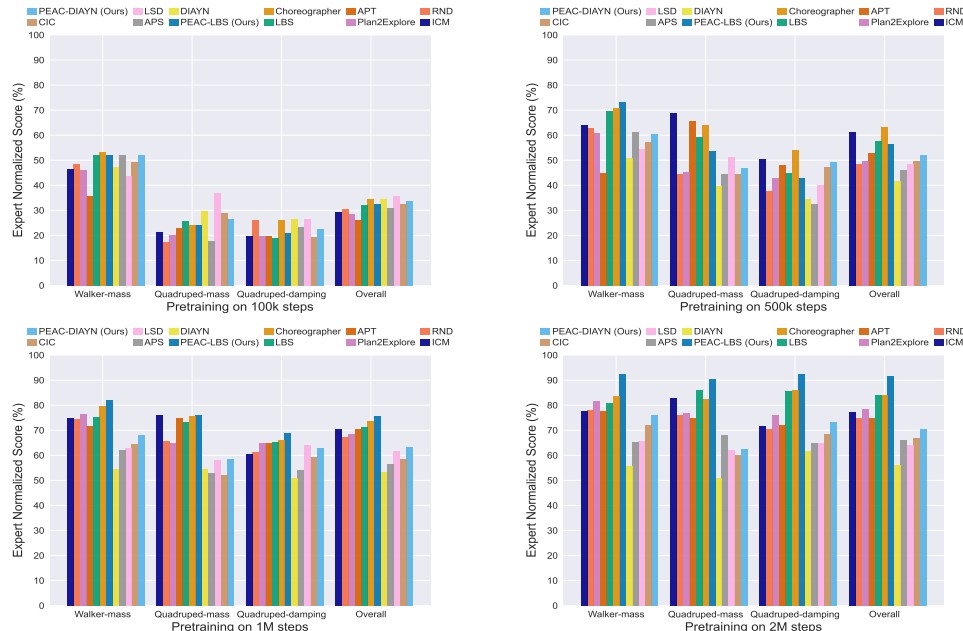

Figure 8: **Ablation study of pre-training steps in image-based DMC**.

### B.8 More Ablation Studies

In this part, we conclude more ablation studies to better clarify the contribution of each component in PEAC. First, we supplement ablation studies of the hyperparameter $\beta$ in Eq. 2 ($\beta$ is set to 1.0 in all our experiments). As discussed in the paper, $\beta$ is a parameter that is negatively related to the fine-tuning timesteps and is for balancing the policy improvement term and the policy constraint term. When the fine-tuning timestep tends to the infinity, $\beta$ tends to 0. Unfortunately, the relationship between $\beta$ and the fine-tuning timesteps is complicated. Thus we evaluate PEAC-LBS under different $\beta$ as below

| Domains | Walker-mass | | | | Quadruped-mass | | | | Quadruped-damping | | | | Normilized |
| Tasks | stand | walk | run | flip | stand | walk | run | jump | stand | walk | run | jump | Average |
|---|---|---|---|---|---|---|---|---|---|---|---|---|---|
| $\beta = 1.0$ | 964.5 | 892.1 | 418.3 | 673.5 | 917.7 | 744.5 | 607.7 | 814.9 | 908.3 | 775.9 | 648.2 | 784.1 | 0.92 |
| $\beta = 0.1$ | 963.8 | 877.6 | 404.8 | 604.4 | 905.3 | 820.0 | 477.5 | 797.0 | 903.0 | 757.6 | 648.3 | 807.6 | 0.90 |
| $\beta = 0.5$ | 958.6 | 896.4 | 416.5 | 640.8 | 929.6 | 794.3 | 593.9 | 806.7 | 921.2 | 746.3 | 540.5 | 794.7 | 0.90 |
| $\beta = 2.0$ | 967.6 | 891.5 | 433.8 | 650.2 | 945.2 | 542.2 | 499.9 | 780.0 | 885.9 | 773.3 | 499.6 | 742.2 | 0.86 |
| $\beta = 3.0$ | 961.7 | 901.6 | 399.6 | 634.9 | 892.7 | 681.0 | 440.1 | 728.1 | 906.0 | 486.9 | 466.9 | 644.5 | 0.81 |

Table 13: **Ablation for $\beta$ of PEAC-LBS in image-based DMC**. Average cumulative reward (mean of 3 seeds) of the best policy trained by different algorithms.

As shown in Table 13, when $\beta$ is large, the performance of PEAC-LBS decreases more than $\beta$ is small, but PEAC-LBS is overall stable with different $\beta$.

Moreover, to clarify the effectiveness of our embodiment discriminator, we supplement LBS-Context and DIAYN-Context, i.e., combining LBS and DIAYN with the embodiment discriminator in PEAC, which utilizes embodiment information during the pre-training stage. Our results in state-based DMC and Image-based DMC are in Table 14 and Table 15, respectively.

| Domains | Walker-mass | | | | Quadruped-mass | | | | Quadruped-damping | | | | Normilized |
| Tasks | stand | walk | run | flip | stand | walk | run | jump | stand | walk | run | jump | Average |
|---|---|---|---|---|---|---|---|---|---|---|---|---|---|
| LBS | 618.0 | 370.3 | 136.8 | 343.1 | 740.8 | 499.1 | 388.7 | 517.2 | 574.0 | 302.0 | 258.4 | 335.8 | 0.50 |
| LBS-Context | 784.3 | 584.7 | 207.6 | 389.0 | 610.1 | 273.7 | 308.1 | 423.1 | 478.3 | 355.8 | 300.4 | 372.2 | 0.52 |
| DIAYN | 502.1 | 245.2 | 106.8 | 212.7 | 682.7 | 484.3 | 371.0 | 469.1 | 553.4 | 386.7 | 331.8 | 394.8 | 0.46 |
| DIAYN-Context | 657.0 | 341.1 | 153.1 | 301.0 | 735.4 | 495.2 | 415.5 | 581.5 | 688.4 | 525.5 | 290.1 | 477.0 | 0.56 |
| PEAC (Ours) | 823.8 | 499.9 | 210.6 | 320.5 | 786.0 | 754.5 | 388.3 | 645.6 | 712.3 | 644.1 | 393.5 | 541.8 | 0.67 |

Table 14: **Ablation study for baselines w/ our embodiment discriminator in state-based DMC**.

| Domains | Walker-mass | | | | Quadruped-mass | | | | Quadruped-damping | | | | Normilized |
| Tasks | stand | walk | run | flip | stand | walk | run | jump | stand | walk | run | jump | Average |
|---|---|---|---|---|---|---|---|---|---|---|---|---|---|
| DIAYN | 772.6 | 515.1 | 193.8 | 365.7 | 583.9 | 425.9 | 311.9 | 431.8 | 791.8 | 410.8 | 367.4 | 536.1 | 0.56 |
| DIAYN-Context | 946.9 | 821.9 | 357.9 | 465.2 | 733.7 | 248.8 | 251.1 | 423.1 | 899.0 | 350.2 | 399.7 | 544.9 | 0.64 |
| PEAC-DIAYN (Ours) | 954.5 | 731.8 | 305.9 | 491.1 | 720.5 | 420.1 | 446.6 | 548.8 | 867.3 | 503.6 | 440.8 | 671.6 | 0.71 |
| LBS | 937.9 | 754.4 | 365.1 | 531.2 | 900.1 | 732.3 | 535.1 | 777.4 | 883.4 | 731.7 | 511.1 | 758.5 | 0.84 |
| LBS-Context | 933.3 | 792.4 | 305.4 | 530.7 | 907.7 | 604.9 | 477.7 | 776.7 | 879.2 | 797.7 | 627.3 | 808.8 | 0.84 |
| PEAC-LBS (Ours) | 964.5 | 892.1 | 418.3 | 673.5 | 917.7 | 744.5 | 607.7 | 814.9 | 908.3 | 775.9 | 648.2 | 784.1 | 0.92 |

Table 15: **Ablation study for baselines w/ our embodiment discriminator in image-based DMC**.

As shown in these two tables, LBS-Context and DIAYN-Context own comparable or superior performance compared with LBS and DIAYN respectively, and PEAC still significantly outperforms them. Consequently, this ablation study highlights that both the embodiment discriminator and cross-embodiment intrinsic rewards $\mathcal{R}_{CE}$ are effective for handling CEURL.

### B.9 Generalization results of pre-trained models

In Fig. 9, we evaluate the generalization ability of pre-trained models to unseen embodiments of all exploration methods in Walker-mass of image-based DMC. After pre-training on several embodiments, we zero-shot utilize these agents to sample trajectories via two different unseen embodiments. Given the trajectories, we reduce the dimension of the hidden states calculated by the world model via t-SNE [57], where points with different colors represent data generated by different embodiments. As shown in Fig. 9, all the baselines can not distinguish different embodiments, while our PEAC-LBS can roughly divide them into two regions, indicating the pre-trained model of PEAC-LBS own strong generalization ability to unseen embodiments.

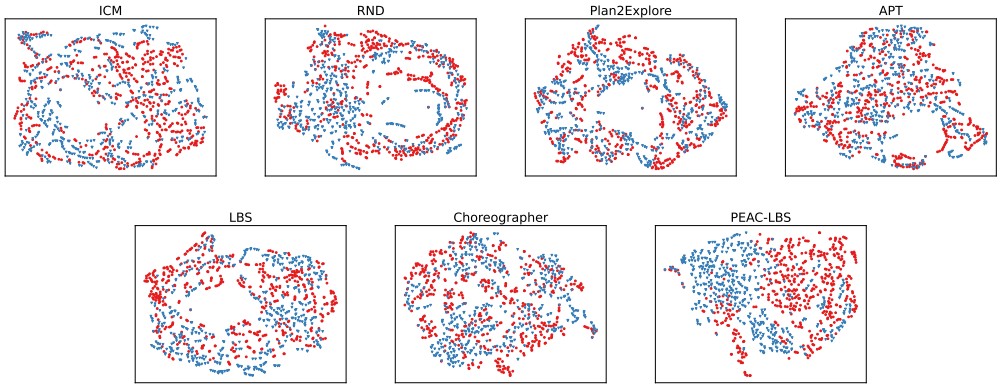

Figure 9: **Visualization of the pre-trained model generalization to unseen embodiments**.

### B.10 Generalization results of fine-tuned models

In this part, we evaluate the generalization ability of the fine-tuned agents to unseen embodiments of state-based DMC and image-based DMC. In these two settings, we pre-train and fine-tune the agent with the sampled training embodiments (Train column in Table 3) and zero-shot evaluate the performance of the fine-tuned agents in the same task but with unseen in-distribution embodiments (Generalization column in Table 3). The detailed generalization results of all downstream tasks in state-based DMC and image-based DMC are in Table 16-17, respectively. As shown in Table 16,

PEAC still significantly outperforms all baselines in normalized average return and there is even a greater leading advantage than baselines, compared with the trained embodiments. This indicates that PEAC can effectively generalize to unseen embodiments and effectively handle downstream tasks.

| Domains | Walker-mass | | | | Quadruped-mass | | | | Quadruped-damping | | | | Normalized |
| Tasks | stand | walk | run | flip | stand | walk | run | jump | stand | walk | run | jump | Average |
|---|---|---|---|---|---|---|---|---|---|---|---|---|---|
| ICM | 702.0 | 467.7 | 146.2 | 246.6 | 321.3 | 165.7 | 158.8 | 258.9 | 259.8 | 112.3 | 135.8 | 134.8 | 0.32 |
| RND | 609.6 | 421.2 | 183.5 | 244.9 | 810.2 | 563.2 | 413.3 | 583.1 | 220.5 | 110.7 | 87.0 | 218.3 | 0.45 |
| Disagreement | 537.6 | 331.6 | 139.8 | 250.0 | 555.7 | 354.7 | 323.4 | 503.2 | 200.3 | 118.4 | 110.0 | 131.4 | 0.36 |
| ProtoRL | 742.1 | 494.0 | 203.9 | 320.5 | 626.5 | 420.9 | 343.2 | 495.7 | 545.4 | 299.1 | 236.6 | 293.1 | 0.51 |
| LBS | 628.0 | 412.0 | 142.0 | 339.4 | 747.7 | 462.1 | 370.7 | 452.4 | 553.8 | 290.4 | 245.6 | 312.0 | 0.49 |
| DIAYN | 497.4 | 257.8 | 107.5 | 207.5 | 677.9 | 402.0 | 366.3 | 451.1 | 547.9 | 361.4 | 328.1 | 387.1 | 0.45 |
| SMM | 680.8 | 561.4 | 232.6 | 315.6 | 309.4 | 144.5 | 171.2 | 244.0 | 278.1 | 116.8 | 115.2 | 211.0 | 0.36 |
| APS | 663.7 | 481.6 | 138.0 | 291.9 | 605.7 | 464.0 | 285.9 | 502.7 | 388.2 | 199.5 | 246.5 | 329.4 | 0.46 |
| CIC | 859.5 | 607.8 | 235.9 | 312.4 | 763.7 | 601.6 | **432.8** | **630.9** | 224.3 | 139.8 | 112.1 | 179.4 | 0.52 |
| BeCL | **874.2** | **693.4** | **255.9** | **354.5** | 683.0 | 369.7 | 349.6 | 517.5 | 522.0 | 425.1 | 285.7 | 491.4 | 0.59 |
| PEAC (Ours) | 860.0 | 554.3 | 225.3 | 324.8 | **776.3** | **741.7** | 381.5 | 624.4 | **734.6** | **641.9** | **385.7** | **537.0** | **0.68** |

Table 16: **Detailed results in state-based DMC in evaluation embodiments**. Average cumulative reward (mean of 10 seeds) of the best policy trained by different algorithms.

Similarly, Table 17 shows that PEAC-LBS not only outperforms baselines but also owns a greater leading advantage than baselines, compared with the trained embodiments. Moreover, PEAC-DIAYN exceeds other pure-exploration methods and demonstrates strong generalization ability.

| Domains | Walker-mass | | | | Quadruped-mass | | | | Quadruped-damping | | | | Normalized |
| Tasks | stand | walk | run | flip | stand | walk | run | jump | stand | walk | run | jump | Average |
|---|---|---|---|---|---|---|---|---|---|---|---|---|---|
| DIAYN | 793.5 | 537.7 | 198.1 | 370.5 | 565.8 | 380.5 | 333.2 | 365.3 | 748.8 | 401.7 | 365.1 | 499.2 | 0.54 |
| APS | 927.8 | 601.8 | 238.6 | 473.2 | 781.6 | 442.2 | 430.2 | 706.3 | 849.9 | 409.8 | 377.1 | 550.4 | 0.67 |
| LSD | 921.4 | 706.9 | 239.4 | 362.4 | 737.2 | 401.8 | 369.2 | 534.6 | 620.6 | 444.2 | 487.3 | 601.6 | 0.63 |
| CIC | 961.5 | 756.1 | 308.9 | 421.4 | 865.3 | 397.1 | 355.1 | 502.8 | 857.1 | 453.1 | 403.6 | 562.3 | 0.67 |
| PEAC-DIAYN (Ours) | 964.1 | 779.1 | 340.3 | 485.7 | 693.5 | 412.1 | 422.3 | 510.8 | 849.0 | 540.3 | 436.8 | 656.6 | 0.70 |
| ICM | 958.3 | 793.8 | 335.6 | 487.9 | 907.5 | 597.0 | 450.2 | 786.2 | 860.4 | 467.9 | 407.9 | 668.1 | 0.77 |
| RND | 963.6 | 825.5 | 360.7 | 506.8 | 843.8 | 483.1 | 429.3 | 743.6 | 841.1 | 449.2 | 407.3 | 714.7 | 0.76 |
| Plan2Explore | 967.8 | 862.4 | 366.2 | 517.8 | 906.0 | 648.6 | 487.5 | 653.2 | 837.2 | 550.9 | 419.8 | 671.1 | 0.78 |
| APT | 938.0 | 811.1 | 357.5 | 467.0 | 820.1 | 485.7 | 484.8 | 689.2 | 777.0 | 526.0 | 431.3 | 645.7 | 0.74 |
| LBS | 944.6 | 789.7 | 387.3 | 529.1 | 898.8 | 696.4 | 542.6 | 765.8 | 875.7 | **770.4** | 524.1 | 761.5 | 0.85 |
| Choreographer | 956.8 | 849.9 | 408.4 | 542.0 | **921.1** | 648.4 | 446.4 | 748.5 | 884.3 | 723.8 | 592.8 | 727.5 | 0.84 |
| PEAC-LBS (Ours) | **967.1** | **902.1** | **444.9** | **695.6** | 901.9 | **750.6** | **598.8** | **799.9** | **897.2** | 748.5 | **659.4** | **798.7** | **0.92** |

Table 17: **Detailed results in image-based DMC in evaluation embodiments**. Average cumulative reward (mean of 3 seeds) of the best policy trained by different algorithms.

## B.11 More challenging tasks and varying embodiments

In this section, we will consider CEURL in much more challenging tasks and more varying embodiment distributions, which are significant future directions for unsupervised cross-embodiment agents in more challenging real-world applications.

We first consider more complicated tasks including locomotion in complicated terrain. Following previous work [16], we design locomotion tasks in incline terrains and the results are below.

| Domains | Walker-mass-incline | | | |
| Task | stand | walk | run | flip |
|---|---|---|---|---|
| LBS | 489.1 | 156.0 | 748.4 | 493.0 |
| PEAC-LBS (Ours) | **557.9** | **245.0** | **748.8** | **681.7** |

Table 18: **Detailed results of Walker-mass-incline in image-based DMC**.

As shown in Table 18, the performance of LBS and PEAC-LBS decreases when locomoting in the incline terrain due to its complexity. PEAC-LBS still significantly outperforms LBS, expressing that our method, especially the cross-embodiment intrinsic rewards, benefits cross-embodiment unsupervised pre-training for handling more complicated tasks.

Besides more complicated tasks, one possible future direction is to consider more different, or even exactly different embodiments. We take the first step by designing several settings with more varying and challenging embodiment distributions:

- Walker-Cheetah: includes two Walker robots with a mass of 0.4 and 1.6 times the normal mass, as well as two Cheetah robots with a mass of 0.4 and 1.6 times the normal mass.

- Walker-Humanoid: includes one Walker robot and one Humanoid robot. Their robot properties, robot shapes, and action spaces are all different.

- Walker-length and Cheetah-torsolength [71]: The former includes walker robots with different foot lengths while the second one includes cheetah robots with different torso lengths. Thus robots' properties and morphologies are different. The figures of these embodiments are in Fig. 10.

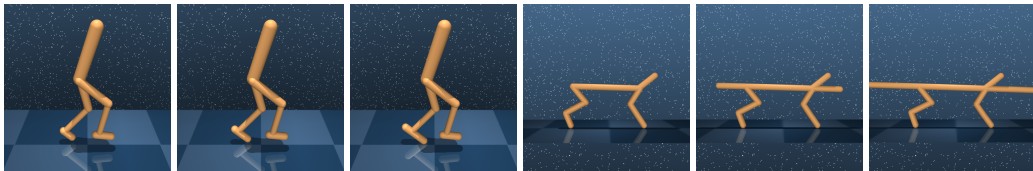

Figure 10: **Benchmark environments** of Walker-length and Cheetah-torsolength. In Walker-length, the **length of the left foot sole** of different robots is different. In Cheetah-torsolength, the **length of the torso** is different.

We mainly compare our PEAC-DIAYN and PEAC-LBS with DIAYN, LBS, and Choreographer in embodiment distributions: Walker-Cheetah, Walker-Humanoid, Walker-length, and Cheetah-torsolength. of which the results are in Table 19, Table 20, and Table 21 respectively.

| Domains Task | Walker-Cheetah | | | |
|---|---|---|---|---|
| | Walker-stand & Cheetah-run | Walker-run & Cheetah-run | Walker-flip & Cheetah-run | Walker-flip & Cheetah-flip |
| DIAYN | 414.3 | 246.2 | 346.6 | 448.3 |
| PEAC-DIAYN (Ours) | 632.6 | 297.5 | 442.2 | 527.5 |
| LBS | 604.8 | 311.7 | 401.0 | 646.2 |
| Choreographer | **681.4** | 374.2 | 446.9 | 624.4 |
| PEAC-LBS (Ours) | **671.2** | **390.7** | **452.3** | **679.2** |

Table 19: **Detailed results of Walker-Cheetah in image-based DMC**.

| Domains Task | Walker-Humanoid | | | | | | | | |
|---|---|---|---|---|---|---|---|---|---|
| | stand-stand | stand-walk | stand-run | walk-stand | walk-walk | walk-run | run-stand | run-walk | run-run |
| DIAYN | 445.1 | 437.7 | 423.7 | 331.5 | 335.9 | 339.3 | 115.3 | 139.3 | 127.0 |
| PEAC-DIAYN (Ours) | 470.4 | 447.2 | 476.3 | 409.6 | 355.6 | 363.1 | 135.6 | 126.9 | 135.9 |
| LBS | **478.9** | **485.2** | 476.3 | **463.6** | 461.0 | 455.3 | 179.9 | 205.6 | 186.2 |
| Choreographer | 471.0 | **479.9** | **483.7** | 409.8 | 413.6 | 403.0 | **216.4** | **233.1** | 160.3 |
| PEAC-LBS (Ours) | 468.4 | **480.3** | **482.3** | 460.8 | 470.5 | 466.1 | 196.8 | **234.4** | **242.8** |

Table 20: **Detailed results of Walker-Humanoid in image-based DMC**.

| Domains Task | Walker-length | | | | Cheetah-torso_length | | | |
|---|---|---|---|---|---|---|---|---|
| | stand | walk | run | flip | run | run_backward | flip | flip_backward |
| DIAYN | 748.5 | 764.0 | 328.7 | 532.3 | 721.1 | **723.6** | 634.2 | 502.4 |
| PEAC-DIAYN (Ours) | **962.7** | **955.9** | **564.3** | **900.6** | 695.4 | 664.5 | 689.2 | 507.8 |
| LBS | **965.7** | 951.9 | 525.7 | 863.4 | 718.4 | 685.4 | 680.6 | 499.5 |
| Choreo | 961.4 | 958.5 | 556.3 | **918.0** | 708.1 | 700.3 | 649.0 | 459.7 |
| PEAC-LBS (Ours) | 966.1 | 956.6 | 573.4 | 899.3 | 731.8 | 704.4 | **747.5** | **515.4** |

Table 21: **Detailed results of Walker-length and Cheetah-torso_length in image-based DMC**.

As shown in these tables, PEAC can achieve much greater performance compared to baselines. These experiments indicate that PEAC has powerful abilities to handle various kinds of embodiment differences, including different morphologies. Unfortunately, when the embodiments vary a lot (like Walker-Humanoid), the performance of PEAC is still limited, thus designing more effective methods for handling complicated embodiments like Humanoid is a promising future direction for further considering cross-embodiment settings.

## B.12 Real-World Applications

As a supplement of Sec. 5.4, we provide more detailed images of real-world robot deployments. As shown in Fig. 11, our method can fast fine-tune to different embodiments and handle different terrains, which are unseen in the simulation. A detailed video is provided on the paper homepage.

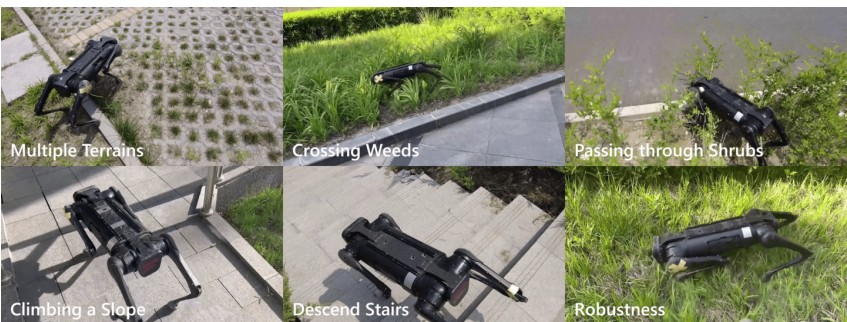

Figure 11: Real-world results for Aliengo robot with different joint failure in different terrains.

## B.13 Computing Resource

In experiments, all the agents are trained by GeForce RTX 2080 Ti with Intel(R) Xeon(R) Silver 4210 CPU @ 2.20GHz. In Image-based DMC / state-based DMC / Robosuite / Isaacgym, pre-training each algorithm (each seed, domain) takes around 2 / 0.5 / 1.5 / 2 days respectively.

## C Pseduo-codes of Algorithms

---

**Algorithm 1** Pre-trained Embodiment-Aware Control (PEAC)

---

**Require:** $M$ training embodiments $\{e_m\}_{m=1}^M$, $M$ replay buffers $\{\mathcal{D}_m\}_{m=1}^M$, $N$ testing embodiments $\{e_{M+n}\}_{n=1}^N$, initialize neural network parameters of the policy

1: // Pre-Training
2: **while** *is unsupervised phase* **do**
3:      // Data Collection
4:      **for** $m = 1, 2, ..., M$ **do**
5:          Sample state-action pairs $\{(s_t^m, a_t^m)\}_t$ with the policy by controlling the embodiment $e_m$ and store them into $\mathcal{D}_m$.
6:      **end for**
7:      // Model Training
8:      **for** update step $= 1, 2, ..., U$ **do**
9:          Sample state-action pairs form each replay buffer $\{(s_t^i, a_t^i)_{t=1}^T\} \sim \mathcal{D}_i, i = 1, 2, ..., M$
10:         Update the embodiment discriminator via these data.
11:         Compute the cross-embodiment intrinsic reward $\mathcal{R}_{\text{CE}}$ for each state-action pair and concatenate them together.
12:         Update the policy by RL backbones (like PPO, DDPG, DreamerV2, and so on) with these data and $\mathcal{R}_{\text{CE}}$.
13:      **end for**
14: **end while**
15: // Fine-Tuning
16: **while** *is supervised phase* **do**
17:      Sample state-action-reward pairs with extrinsic rewards $\mathcal{R}_{\text{ext}}$ via embodiment $e_m$ and store them into $\mathcal{D}_m$.
18:      Update the policy by jointly training data from different replay buffers via RL backbones.
19: **end while**
20: // Evaluation
21: Evaluate fine-tuned policy with downstream task $\mathcal{R}_{\text{ext}}$ via $\{e_m\}_{m=1}^M$ and unseen embodiments $\{e_{M+n}\}_{n=1}^N$.

---

---

**Algorithm 2** PEAC-LBS

---

**Require:** $M$ training embodiments $\{e_m\}_{m=1}^M$, $M$ replay buffers $\{\mathcal{D}_m\}_{m=1}^M$, $N$ testing embodiments $\{e_{M+n}\}_{n=1}^N$, initialize neural network parameters of the policy
1: // Pre-Training
2: **while** *is unsupervised phase* **do**
3:     // Data Collection (the same as PEAC)
4:     ...
5:     // Model Training
6:     **for** update step $= 1, 2, ..., U$ **do**
7:         Sample state-action pairs form each replay buffer $\{(s_t^i, a_t^i)_{t=1}^T\} \sim \mathcal{D}_i, i = 1, 2, ..., M$
8:         Update the embodiment discriminator via these data.
9:         Update the components of LBS, including the Latent Prior model, the Latent Posterior model, and the Reconstruction model (In DreamerV2 backbone, we can directly utilize its prior model and posterior model).
10:        Compute the intrinsic reward $\mathcal{R}_{\mathrm{CE}} + \mathcal{R}_{\mathrm{LBS}}$ for each state-action pair and concatenate them together.
11:        Update the policy by RL backbones (like PPO, DDPG, DreamerV2, and so on) with these data and $\mathcal{R}_{\mathrm{CE}} + \mathcal{R}_{\mathrm{LBS}}$.
12:     **end for**
13: **end while**
14: // Fine-Tuning(the same as PEAC)
15: ...
16: // Evaluation
17: Evaluate fine-tuned policy with downstream task $\mathcal{R}_{\mathrm{ext}}$ via $\{e_m\}_{m=1}^M$ and unseen embodiments $\{e_{M+n}\}_{n=1}^N$.

---

---

**Algorithm 3** PEAC-DIAYN

---

**Require:** $M$ training embodiments $\{e_m\}_{m=1}^M$, $M$ replay buffers $\{\mathcal{D}_m\}_{m=1}^M$, $N$ testing embodiments $\{e_{M+n}\}_{n=1}^N$, initialize neural network parameters of the behavior policy conditioned on skill and embodiment context $\pi(\cdot|s, z, e)$, initialize neural network parameters of the embodiment-aware skill policy $\pi(z|e, \tau)$
1: // Pre-Training
2: **while** *is unsupervised phase* **do**
3:     // Data Collection (the same as PEAC)
4:     ...
5:     // Model Training
6:     **for** update step $= 1, 2, ..., U$ **do**
7:         Sample state-action pairs form each replay buffer $\{(s_t^i, a_t^i)_{t=1}^T\} \sim \mathcal{D}_i, i = 1, 2, ..., M$
8:         Update the embodiment discriminator via these data.
9:         Update the skill discriminator of DIAYN via these data.
10:        Compute the intrinsic reward $\mathcal{R}_{\mathrm{CE}} + \mathcal{R}_{\mathrm{DIAYN}}$ for each state-action pair and concatenate them together.
11:        Update the behavior policy conditioned on skill and embodiment context by RL backbones (like PPO, DDPG, DreamerV2, and so on) with these data and $\mathcal{R}_{\mathrm{CE}} + \mathcal{R}_{\mathrm{DIAYN}}$.
12:     **end for**
13: **end while**
14: // Fine-Tuning
15: **while** *is supervised phase* **do**
16:     Sample state-action-reward pairs with extrinsic rewards $\mathcal{R}_{\mathrm{ext}}$ via embodiment $e_m$ and store them into $\mathcal{D}_m$.
17:     Update the embodiment-aware skill policy by jointly training data from different replay buffers via RL backbones.
18: **end while**
19: // Evaluation
20: Evaluate fine-tuned agents with downstream task $\mathcal{R}_{\mathrm{ext}}$ via $\{e_m\}_{m=1}^M$ and unseen embodiments $\{e_{M+n}\}_{n=1}^N$.

---

## D Broader Impact

Designing generalizable agents for varying tasks and embodiments is a major concern in reinforcement learning. This work focuses on cross-embodiment unsupervised reinforcement learning and proposes a novel algorithm PEAC, which leverages trajectories from different embodiments for pre-training, subsequently broadly enhancing performance on downstream tasks. Such advancements provide the potential for future real-world cross-embodiment control. One of the potential negative impacts is that algorithms using deep neural networks, which lack interoperability and face security and robustness issues. There are no serious ethical issues as this is basic research.

