# OpenReview forum: "PEAC: Unsupervised Pre-training for Cross-Embodiment Reinforcement Learning"
_NeurIPS.cc/2024/Conference — NeurIPS 2024 poster_

### Official Review · Reviewer_RG3H · 2024-07-10

**Soundness:** 3
**Presentation:** 3
**Contribution:** 3
**Rating:** 6
**Confidence:** 4

**Summary:**

This paper presents a new unsupervised reinforcement learning (URL) algorithm called Pre-trained Embodiment-Aware Control (PEAC). PEAC is designed to tackle cross-embodiment tasks, explicitly considering the influence of different embodiments to facilitate exploration and skill discovery across embodiments. Experimental results demonstrate that PEAC notably enhances adaptation performance.

**Strengths:**

1. The paper is well-written and easy to follow. The intuitive figures and logically structured text make PEAC clear and understandable.
2. The author(s) conduct comprehensive experiments and present a solid work.
3. As embodied intelligence gains increasing attention, cross-embodiment research is indeed an important direction. PEAC undoubtedly contributes to the community in this regard.

**Weaknesses:**

1. Although the author(s) compare various standard and SOTA baselines in the experiments, they lack strong persuasiveness. Since the author(s) focus on cross-embodiment tasks, it is apparent that the baselines, which do not utilize embodiment information $e$, exhibit poorer performance. Therefore, it is unclear whether PEAC's high performance is due to the use of embodiment information $e$ or the utilization of cross-embodiment intrinsic rewards $\mathcal R_{\text{CE}}$. To highlight the paper's contribution, I suggest that the author(s) modify some baselines to naively incorporate embodiment information and then compare with them. For example, conditioning the policy and reward of DIAYN on embodiment $e$, etc.
2. Experiments conducted on Robosuite effectively showcase "cross-embodiment," while other environments may not do so as prominently. By modifying parameters like "mass" and "damping" in DMC tasks or introducing joint torque failures in Unitree A1, the setting becomes more like a Meta RL, where the robot's morphology remains relatively unchanged. In locomotion tasks, "cross-embodiment" could be exemplified by robots like Gym-MuJoCo's Walker2d, DMC's Walker, and Humanoid, where they all use two legs for alternating walking and aim to enhance mobility. However, they differ in morphology. Demonstrating whether PEAC can learn the same shared knowledge of bipedal walking from these diverse morphologies may better illustrate the topic of "cross-embodiment."

**Questions:**

Please see Weaknesses.

**Limitations:**

My primary concern is that the baselines in the experiments do not utilize embodiment information $e$. Please refer to the first point in Weaknesses.

---

> ### Author Rebuttal · Authors · 2024-08-06
>
> Thanks a lot for your review, especially the praise for PEAC's contributions to embodied intelligence. Below we will address all your concerns.
>
> **W1:** About baselines utilizing embodiment information $e$.
>
> **A:** Thank you for recognizing that we have included various standard and SOTA baselines. Following your valuable suggestions of adding baselines incorporating embodiment information, we have supplemented experiments of **LBS-Context** and **DIAYN-Context**, which inherit LBS and DIAYN by incorporating embodiment information $e$, to better highlight the paper's contribution. Our results are below:
>
> | State-based DMC | Walker-mass | Quadruped-mass | Quadruped-damping |
> |-|-|-|-|
> | DIAYN  | 266.7  | 501.8| 416.7 |
> | DIAYN-context  | 364.4  | 564.2 | 499.5  |
> | LBS | 367.1 | 536.5 | 367.6 |
> | LBS-context | 365.9 | 514.5  | 417.4|
> | PEAC  | **463.7** | **643.6** | **572.9** |
>
> | Image-based DMC | Walker-mass | Quadruped-mass | Quadruped-damping |
> |-|-|-|-|
> | DIAYN | 461.8 | 438.4  | 526.5  |
> | DIAYN-context  | 572.0 | 457.6 | 546.6 |
> | PEAC-DIAYN | 587.3 | 578.4 | 572.8 |
> | LBS | 647.2 | 736.2 | 721.2 |
> | LBS-context | 655.1  | 739.1 | 722.5  |
> | PEAC-LBS | **727.6** | **755.2** | **740.2** |
>
> As shown in these two tables, LBS-Context and DIAYN-Context own superior performance compared with LBS and DIAYN respectively, and PEAC still significantly outperforms these baselines. Especially, in state-based DMC, the improvement in the performance of LBS-Context and DIAYN-Context is more remarkable than in image-based DMC. It may be because, in image-based DMC, we take DreamerV2 as the RL backbone, which has encoded historical information for making the decision and is possible to distinguish different embodiments through historical information to some degree. But in state-based DMC, we take DDPG as the RL backbones following previous works [1], which are Markovian policies and thus utilizing embodiment information during the pre-training is helpful (Lines 210-216). Consequently, our supplemented results **highlight that both the embodiment discriminator and cross-embodiment intrinsic rewards $\mathcal{R}_{\text{CE}}$ are effective for handling CEURL** (more details, including figures with aggregate metrics of these methods, are in the global response and the corresponding PDF).
>
> **W2:** About experimental settings that can better highlight cross-embodiment.
>
> **A:** Thank you for your recognition of our experiments on Robosuite. In DMC and Isaacgym, we mainly modify the property parameters or introduce joint torque failures because embodiments with different properties are **one of the most basic cross-embodiment settings**, which is worth paying attention to especially when we first research CEURL. Moreover, although these embodiments only modify property parameters, **their ability to handle tasks may differ a lot**, e.g. in our real-world experiments, when the front foot joint of the A1 robot fails, it can only drag its legs forward, while when the rear foot joint fails, it will lift its rear legs to move forward (Fig. 7 and video in the supplementary material).
>
> In experiments of DMC, besides settings with different properties like Walker-mass, we have also included settings with different morphology in **Appendix B.10**: **Walker-Cheetah**, where Walker and Cheetah both own two legs but their morphology differs a lot. Experiments in Appendix B.10 demonstrate that PEAC can also significantly outperform baselines and reveal PEAC's potential in handling exactly different embodiments.
>
> Moreover, following your constructive suggestions, we have also supplemented extensive experiments in DMC considering settings with greater differences across embodiments, **especially differing in morphology**:
>
> - **Walker-Humanoid**: Following your constructive suggestions, we consider the cross-embodiment setting that includes Walker robots and Humanoid robots.  Their robot properties, robot shapes, and action spaces are all different.
> - **Walker-length** and **Cheetah-torsolength** [2]: The former includes walker robots with different foot lengths while the second one includes cheetah robots with different torso lengths. Thus robots' properties and morphologies are different  (Figs of these embodiments are in the attached PDF of the global response).
>
> The results in these settings are below:
>
> | Walker-Humanoid | stand-stand | stand-walk | stand-run | walk-stand | walk-walk | walk-run | run-stand | run-walk | run-run | average      |
> |-|-|-|-|-|-|-|-|-|-|-|
> | DIAYN  | 286.8 | 320.2  | 307.2 | 185.4 | 175.4 | 209.7 | 75.4  | 61.1 | 55.4 | 186.3  |
> | PEAC-DIAYN | 231.6 | 430.3  | 371.7  | 247.4 | 210.7 | 245.2 | 88.3 | 85.1 | 69.9  | 220.0  |
> | LBS | 477.8 | 463.2   | 475.7 | **407.1** | 343.7 | 401.1 | 150.8  | 112.3 | 134.6 | 329.6  |
> | PEAC-LBS | **486.4** | **484.9**  | **483.6**  | **406.9**   | **373.6**  | **432.5** | **180.4** | **145.7** | **188.2** | **353.6**  |
>
> | Walker-length | stand | walk | run| flip | average  |
> |-|-|-|-|-|-|
> | LBS | **977.5** | 909.1 | **557.5** | 630.4 | 768.6 |
> | PEAC-LBS | **970.0**  | **976.4** | 544.2 | **764.9** | **813.9**  |
>
> | Cheetah-torsolength | run | run_backward | flip  | flip_backward | average  |
> |-|-|-|-|-|-|
> | LBS | 625.9 | **512.9** | 611.9 | 523.0 | 568.4  |
> | PEAC-LBS| **745.3** | 499.7 | **646.4** | **649.2** | **635.2** |
>
> As shown in these tables, PEAC can achieve much greater performance compared to baselines. These experiments indicate that PEAC has powerful abilities to handle various kinds of embodiment differences, including different morphologies. Also, designing more effective methods for handling complicated embodiments like Humanoid is a promising future direction.
>
> Reference:
>
> [1] URLB: Unsupervised Reinforcement Learning Benchmark
>
> [2] Learning Robust State Abstractions for Hidden-Parameter Block MDPs

---

> > ### Comment · Reviewer_RG3H · 2024-08-08
> >
> > Thank you very much for the thorough response. The additional experiments have completely addressed my concerns, and I will raise the rating by 2 points.

---

> ### Author Response · Authors · 2024-08-08
> **Thanks a lot for your feedback**
>
> Dear reviewer RG3H,
>
> Thank you very much for your valuable suggestions and increasing the score. We will try our best to further improve our paper.
>
> best regards,
>
> Authors

---

### Official Review · Reviewer_xr64 · 2024-07-12

**Soundness:** 3
**Presentation:** 2
**Contribution:** 3
**Rating:** 6
**Confidence:** 3

**Summary:**

This papaer introduces a novel setting, cross-emodiment unsupervised RL, which deals with pre-training good policies in reward free environments across different embodiments in order to perform well on downstream tasks on unseen embodiments. The authors propose the algorithm PEAC for unsupervised learning in this setting. Their method proposes a novel intrinsic reward which is derived from the minmax objective of maximizing the improvement in the downstream fine-tuned policy over the pre-trained policy minus a regularizing KL term, which ensures that the fine-tuned policy stays close to the pre-trained policy, under the worst-case extrinsic reward. From this objective, they derive an intrinsic reward for the unsupervised cross-embodiment setting with maximizes the (fixed) log prior over embodiments minus the (learned) log posterior over embodiments, which encourages the agent to seek areas of the state space that are common accross embodiments, or underexplored. They conduct extensive experiments in both simulated and real environments, including both state-based and image-based DMC tasks, and compare against a large number of pre-existing skill-based and exploratoin based unsupervised RL methods. Their results are impressive, showing improvement in almost all cases over the baselines. Moreover, they conduct real-world experiments in which their algorithm similarly outperforms.

**Strengths:**

- Introduces a novel setting: cross-embodiment unsupervised RL
- Convincing motivation - the idea that most cross-embodiment training which focuses on a single task may not learn generalizable embodiment-specific skills rather than task-specific skills
- Extensive experiments in both simulation and real-world settings with very strong results compared to baselines

**Weaknesses:**

- The paper is hard to follow and key details on the implementation are not fully explained. For example in line 68 a "embodiment context encoder" is mentioned but I do not see this explained any where else in the paper. How is the discriminator trained?
- The extension of the information geometry analyses in Sec 3 do not seem particularly related to the main claims of the paper. This section serves primarly to highlight that policies may be stochastic, however it does not seem that the this information is utilized in the algorithm presented
- The stated goal of the paper is to better learn embodiement-aware skills, but the policy is optimized to maximize the expectation over all embodiments. Would this encourage embodiment-aware skills or rather embodiment-agnostic skills?

**Questions:**

1. It is not clear to me how the cross-embodiment policy handles different state spaces. For example in the cheetah-walker experiments in the appendix, how is the policy architecture designed such that it can take in different size state vectors?
2. What does super script c in the MDP M represent?
3. Can you please explain the equality in the last line of Eq 29?
4. It is not clear what is unique about the cross-embodiment setting for unsupervised RL versus the more generic contextual MDP setting. Why do embodiments needed to be treated differently?
5. If log(p(e)) is fixed, does it need to be in the reward?

**Limitations:**

- As the authors note, their experiments are limited to settings where the embodiments are very similar. It is not clear how their method extends to dissimilar embodiments, particularly with different sized state vectors.

---

> ### Author Rebuttal · Authors · 2024-08-06
>
> Thanks a lot for the supportive review and constructive suggestions. Below we address the detailed comments.
>
> **W1:** About the embodiment context encoder in line 68.
>
> **A:** The embodiment context encoder is the embodiment discriminator in the main text of the paper, and we will polish our paper to make it consistent in context. As for the training of the discriminator, we utilize cross-entropy loss of the one-hot embodiment vector during the pre-training stage, following previous works [1]. We will clarify it in the revised versions.
>
> To better clarify the impact of the embodiment discriminator and the cross-embodiment intrinsic reward $R_{\text{CE}}$, we have supplemented ablation studies of **LBS-Context** and **DIAYN-Context**, i.e., combining LBS and DIAYN with our embodiment discriminator in PEAC but without $R_{\text{CE}}$. Our results are as below:
>
> |State-based DMC| Walker-mass|Quadruped-mass | Quadruped-damping |
> |-|-|-|-|
> |DIAYN|266.7| 501.8| 416.7|
> |DIAYN-context| 364.4|564.2| 499.5|
> |LBS |367.1|536.5| 367.6|
> |LBS-context | 365.9 | 514.5| 417.4|
> |PEAC| **463.7** | **643.6** | **572.9** |
>
> |Image-based DMC|Walker-mass | Quadruped-mass | Quadruped-damping |
> |-|-|-|-|
> |DIAYN|461.8| 438.4| 526.5  |
> |DIAYN-context| 572.0| 457.6 | 546.6 |
> |PEAC-DIAYN | 587.3| 578.4 | 572.8 |
> |LBS| 647.2|736.2|721.2 |
> |LBS-context|655.1|739.1 | 722.5|
> |PEAC-LBS|**727.6**|**755.2**|**740.2**|
>
> As shown in these two tables, LBS-Context and DIAYN-Context own superior performance compared with LBS and DIAYN respectively, and PEAC still significantly outperforms these baselines. Consequently, this ablation study **highlights that both the embodiment discriminator and cross-embodiment intrinsic rewards $R_{\text{CE}}$ are effective for handling CEURL** (more details and results are in the global response and the attached PDF).
>
> **W2:** About the information geometry analyses in Sec. 3.
>
> **A:** Yes, our extension of the information geometry analyses does not directly guide the design of PEAC. As the major contributions of this paper include the novel setting **CEURL** and our algorithm **PEAC**, **our information geometry analyses are mainly related to CEURL** by revealing its difficulty. Moreover, as you have mentioned, our information geometry analyses highlight that policies may be stochastic or history-related, thus our PEAC chooses to encode historical state-action pairs to the embodiment context in practice (Lines 213-216). We will emphasize it in the revised version.
>
> **W3:** About embodiment-aware or embodiment-agnostic skills?
>
> **A:** The goal of **embodiment-aware skill discovery** (Lines 232-252) is to learn similar skills for all these embodiments during the pre-training stage. However, as different embodiments own different properties/morphologies, their actions for achieving similar skills may differ a lot. Thus we name our method embodiment-aware skill discovery. Thanks again for your question, we will discuss the concept more clearly in the revised version.
>
> **Q1:** How to handle different action/state spaces?
>
> **A:** When handling state/action spaces with different dimensions, we will take the maximal dimension of these spaces $d$ and pad the states/actions into $d$ dimension by filling in zeros, following previous works [2].
>
> **Q2:** What does $c$ in the MDP $\mathcal{M}^c$ represent?
>
> **A:** We use $\mathcal{M}$ to represent original MDP, $\mathcal{M}_e$ represent MDP with embodiment $e$, **$\mathcal{M}^c$ represent controlled MDP**, i.e., an MDP without rewards in the pre-training stage, and $\mathcal{M}_e^c$ represent controlled MDP with embodiment $e$.
>
> **Q3:** About the equality in the last line of Eq. 29.
>
> **A:** The last equality of Eq. 29 can be derived by the property of conditional probability and we abbreviate $\mathcal{M}_e^c$ as $e$:
>
> $$\log \frac{p_{\pi}(\tau)}{p_{\pi}(\tau|e)} = \log \frac{p_{\pi}(\tau)}{p_{\pi}(\tau, e) / p_{\pi}(e)} = \log \frac{p_{\pi}(e) }{p_{\pi}(\tau, e) / p_{\pi}(\tau)} = \log \frac{p(e)}{p_{\pi}(e|\tau)}.$$
>
> Thanks for your question, we will provide the detailed derivation in the revised versions.
>
> **Q4:** About the connection between CEURL and contextual MDP.
>
> **A:** As we have mentioned in Lines 125-126, cross-embodiment RL can be formulated by contextual MDP. Moreover, as the cross-embodiment setting is practical for real-world applications and owns several unique properties, it introduces various new problems like cross-embodiment transfer [3] and has received widespread attention in embodied intelligence [3-5] (as mentioned by Reviewer RG3H). In this work, considering that different embodiments may own similar structures and learn similar skills, we propose unsupervised pre-train across embodiments to learn knowledge only specialized in these embodiments themselves, i.e., CEURL, which is a novel setting that existing contextual MDP can not cover and thus we combine contextual MDP with controlled MDP as our CEMDP to formulate our problem.
>
> **Q5:** Does $\log p(e)$ used in the reward?
>
> **A:** No, as we have mentioned in Lines 205-206, our cross-embodiment intrinsic reward eliminates $\log p(e)$ as it is fixed.
>
> **Limitation:** About experimental settings that can better highlight cross-embodiment
>
> **A:**  We have supplemented more cross-embodiments with different morphologies in DMC: Walker-Humanoid, Walker-length, and Cheetah-torsolength. **More details and results are in the global response, where PEAC shows greater performance compared to baselines**. The results indicate that PEAC has powerful abilities to handle various kinds of embodiment differences, including different morphologies.
>
> Reference:
>
> [1] Multi-task reinforcement learning with soft modularization
>
> [2] Learning to Modulate pre-trained Models in RL
>
> [3] Cross-Embodiment Robot Manipulation Skill Transfer using Latent Space Alignment
>
> [4] Xirl: Cross-embodiment inverse reinforcement learning
>
> [5] Pushing the limits of cross-embodiment learning for manipulation and navigation

---

> > ### Author Response · Authors · 2024-08-12
> > **Look forward to further feedback**
> >
> > Dear Reviewer xr64:
> >
> > We sincerely thank you again for your constructive feedback and suggestions. We have tried our best to answer the concerns raised, especially including more ablation studies to explain the embodiment context encoder, and are happy to clarify/discuss any further questions. We hope you may find our response satisfactory and raise your rating accordingly. We are looking forward to hearing from you about any further feedback.
> >
> > Best, Authors.

---

> ### Comment · Area_Chair_YC1z · 2024-08-12
> **Reviewer Discussion Needed**
>
> Dear Reviewer,
>
> The discussion time is coming to an end soon. Please engage in the discussion process which is important to ensure a smooth and fruitful review process. Give notes on what parts of the reviewers responses that have and have not addressed your concerns.

---

> ### Comment · Reviewer_xr64 · 2024-08-12
>
> Dear authors, thank you for the thorough response and for addressing my questions. The additional experiments on more complicated cross-embodiment settings have alleviated some of my concerns. However, I believe the setting is still very similar to multi-task or meta-RL and it is not clear to me why the problem must be treated as distinct except in the treatment of different state and action spaces and, in this respect, the method of zero-padding is slightly underwhelming. That said, the new experiments certainly make the paper's claims more convincing. I will raise my rating by 1 point.

---

> ### Author Response · Authors · 2024-08-12
> **Thanks a lot for your feedback and supportive comments**
>
> Dear Reviewer xr64:
>
> Thank you very much for your constructive feedback and for increasing the score. Below we will answer your questions, especially about the difference between CEURL and multi-task/meta RL.
>
> First, we briefly introduce multi-task/meta RL and CEURL:
>
> - **Multi-task RL**: train an agent to handle several different tasks, represented by MDPs **with rewards**, at the same time.
> - **Meta RL**: pre-train an agent in several training tasks, represented by MDPs **with rewards**. The pre-trained agent is required to fast adapt to testing tasks. Here training tasks and testing tasks are sampled from the same distribution [1,2].
> - **CEURL**: pre-train an agent with several embodiments **without rewards**. The pre-trained agent is required to fast adapt to any testing tasks. Here we have no prior knowledge of the testing task. (lines 137-146)
>
> Consequently, from a modeling perspective, the biggest difference is that **CEURL pre-trains in an unsupervised manner**, i.e., without any extrinsic rewards, but **meta-RL pre-trains with extrinsic rewards**, which obeys the same distribution of the testing tasks.
>
> Moreover, from the concept perspective, multi-task/meta RL regards the embodiment and the task as a whole and utilizes an MDP to handle them together. Differently, CEURL distinguishes between the concept of embodiments and tasks. Thus CEURL hopes to learn embodiment-aware and task-agnostic knowledge by pre-training agents only with these embodiments without any tasks (lines 125-136). This knowledge is considered very helpful, especially when embodied intelligence requires handling different tasks across embodiments in the real world.
>
> Furthermore, we agree that zero-padding is a feasible but not the best method for handling different state/action spaces. And we believe that developing unified action/state embedding spaces is a promising future direction for handling the cross-embodiment setting.
>
> Thanks again for your kind comments and we will add these discussions into the paper to further improve it. Also, we are looking forward to hearing from you about any further feedback.
>
> best regards,
>
> Authors
>
> Reference:
>
> [1] Model-Agnostic Meta-Learning for Fast Adaptation of Deep Networks
>
> [2] Efficient Off-Policy Meta-Reinforcement Learning via Probabilistic Context Variables

---

### Official Review · Reviewer_3BHi · 2024-07-12

**Soundness:** 3
**Presentation:** 3
**Contribution:** 3
**Rating:** 5
**Confidence:** 3

**Summary:**

This paper addresses the challenge of designing generalizable agents capable of adapting to diverse embodiments. The authors propose the CEURL setting as a novel framework for this problem and introduce the PEAC algorithm to address it. Recognizing that CEURL requires minimizing across different downstream tasks while maximizing the fine-tuned policy, PEAC tackles the issue through policy improvement and policy constraint mechanisms. PEAC can integrate with existing unsupervised reinforcement learning methods designed for single-embodiment settings, such as LBS and DIAYN. Through experiments conducted in both simulation and real-world settings, the effectiveness of PEAC is demonstrated.

**Strengths:**

Originality: This paper introduces a novel problem formulation, CEURL, and presents a new algorithm, PEAC.

Quality:
1. The authors thoroughly discuss the challenges inherent in the CEURL problem definition, supported by mathematical proofs. This elevates CEURL from a simple definition to a high-quality problem formulation.
2.Convincing experiments involving real robots are presented.

Clarity: The paper is well-written and easy to follow.

Significance: This paper has the potential to significantly impact cross-embodiment control through its introduction of a new problem formulation and an innovative unsupervised pretraining framework.

**Weaknesses:**

1. Among all the experiments presented, tasks for DMC and Issacgym are relatively limited, with only simple movements. I recommend adding more complex tasks like [1].
2. In ablation study, the authors only show the results of different training timesteps. More ablation studies can help demonstrate how the algorithm works.
3. From my perspective, this paper is closely related to [2], which proposes an unsupervised skill discovery method through contrastive learning. While [2] focuses on single-embodiment scenarios, PEAC extends this approach to multiple embodiments, addressing the additional challenges posed by varying state and action spaces via introducing additional embodiment prior and posterior probabilities.

[1] Gupta, Agrim, et al. "Embodied intelligence via learning and evolution." Nature communications 12.1 (2021): 5721.
[2] Yang, Rushuai, et al. "Behavior contrastive learning for unsupervised skill discovery." International Conference on Machine Learning. PMLR, 2023.

**Questions:**

1. For objective function (2), can you provide an ablation study using only the Policy Improvement part and only the Policy Constraint part?
2. In Figure 6 (right), PEAC-DIAYN shows the worst performance at 100k training steps. What is the cause?
3. In Appendix B.6, the performance of PEAC in Robosuite is fair comparing to that in DMC or Issacgym. Does PEAC work better for simple tasks (like walk, stand) than the tasks in Robosuite?

**Limitations:**

Yes

---

> ### Author Rebuttal · Authors · 2024-08-06
>
> Thank you for your supportive review and valuable suggestions. Below we address the detailed comments for all your questions.
>
> **W1:** About more complex tasks.
>
> **A:** In our experiments, tasks chosen in DMC and Isaacgym are **standard** and **widely used in unsupervised RL evaluation** [1, 2, 3]. These tasks are basic and important to evaluate agents' locomotion ability, and they can also combine to handle much more complex tasks like parkour [2]. Thanks for your suggestion and reference, referring to [4] including tasks of locomotion in complicated terrain like inclines, we have designed more complicated tasks for Walker-mass in DMC to evaluate their locomotion ability in complicated terrain like inclines: **Walker-mass-incline**. The results are shown below:
>
> | Walker-mass-incline | stand |walk|run| flip| average|
> |-|-|-|-|-|-|
> | LBS | 725.2 | 597.7 | 109.3 | 430.6 | 465.7|
> | PEAC-LBS | **793.8** | **670.3** | **268.9** | **509.1** | **560.5** |
>
> As shown in this table, the performance of LBS and PEAC-LBS decreases when locomoting in the incline terrain due to its complexity. **PEAC-LBS still significantly outperforms LBS**, expressing that our method, especially the cross-embodiment intrinsic rewards, benefits cross-embodiment unsupervised pre-training for handling more complicated tasks.
>
> **W2 \& Q1:** More ablation studies, like only using one of the two parts in Eq. 2.
>
> **A:** Thanks for your suggestion. There are two terms in Eq.2: the policy improvement part and the policy constraint part, representing that in the fine-tuning stage, we need to **maximize the policy performance** within **limited training steps**, respectively (Lines 182-188 in the paper). Thus they need to be considered together and it is unreasonable to use only one of these two parts for this setting. Moreover, we introduce a trade-off parameter $\beta$ in Eq.2 to **balance these two terms**, **which is set as 1.0 in all our experiments**. Consequently, we have supplemented ablation of $\beta$, of which the results are:
>
> |Ablation of $\beta$|Walker-mass|Quadruped-mass|
> |-|-|-|
> |PEAC-LBS($\beta$=1.0, default)| **727.6** | **755.2** |
> |PEAC-LBS($\beta$=0.5)|**726.9** |728.8|
> |PEAC-LBS($\beta$=2.0)|713.2|**750.1**|
>
> As shown in this table, when $\beta$ is smaller, the performance of PEAC-LBS decreases a little, and overall PEAC-LBS is stable with different $\beta$.
>
> Besides $\beta$, we have also supplemented more ablation studies, especially about our cross-embodiment intrinsic rewards $R_{\text{CE}}$, we consider **LBS-Context** and **DIAYN-Context** i.e., combining LBS and DIAYN with our embodiment discriminator in PEAC but without $R_{\text{CE}}$. Our results are as below:
>
> | State-based DMC | Walker-mass | Quadruped-mass | Quadruped-damping |
> |-|-|-|-|
> | DIAYN  | 266.7  | 501.8| 416.7|
> | DIAYN-context  | 364.4  | 564.2 | 499.5|
> | LBS | 367.1 | 536.5 | 367.6 |
> | LBS-context | 365.9 | 514.5  | 417.4|
> | PEAC  | **463.7** | **643.6** | **572.9**|
>
> | Image-based DMC | Walker-mass | Quadruped-mass | Quadruped-damping |
> |-|-|-|-|
> | DIAYN | 461.8 | 438.4  | 526.5  |
> | DIAYN-context  | 572.0 | 457.6 | 546.6 |
> | PEAC-DIAYN | 587.3 | 578.4 | 572.8 |
> | LBS | 647.2 | 736.2 | 721.2 |
> | LBS-context | 655.1  | 739.1 | 722.5  |
> | PEAC-LBS | **727.6** | **755.2** | **740.2** |
>
> As shown in these two tables, LBS-Context and DIAYN-Context own superior performance compared with LBS and DIAYN respectively, and PEAC still significantly outperforms these baselines. Consequently, this ablation study **highlights that both the embodiment discriminator and cross-embodiment intrinsic rewards $R_{\text{CE}}$ are necessary for handling CEURL** (more details of results are in the global response).
>
> **W3:** About the relation with the BeCL [2].
>
> **A:** BeCL [2] mainly focuses on encouraging the agent to learn diverse skills related to corresponding behaviors via applying contrastive learning for single-embodiment skill discovery. As you have mentioned, our PEAC considers the cross-embodiment setting and thus is **orthogonal to these skill-discovery methods**.
>
> We have discussed and compared BeCL as well as other skill discovery methods in the paper. Especially, as shown in Sec.4, we have discussed the relationship between the objective of skill discovery (including BeCL) and our cross-embodiment objective, with a result of **a unified cross-embodiment skill-based adaptation objective** (Eq. 6-7). Thus our PEAC can naturally combine with these skill-discovery methods, including BeCL, and we propose PEAC-DIAYN as an example. Thanks again for your question, applying contrastive learning methods like BeCL to our PEAC for handling CEURL is indeed an interesting future direction.
>
> **Q2:** About the performance of PEAC-DIAYN in 100k pre-training steps.
>
> **A:** Thanks for your question. When the pre-training steps are small (like 100k), it is difficult to learn useful knowledge during the pre-training stage (like world models or our embodiment discriminator), and the uncertainty of results is relatively high, thus current research mainly considers the fine-tuning results after pre-training 2M in DMC [1-2].
>
> **Q3:** About the performance of PEAC in Robosuite.
>
> **A:** In Robosuite, the overall performance of PEAC is still 10\% higher than all baselines, indicating the effectiveness of PEAC. Compared with DMC, the morphologies in Robosuite vary a lot more, thus baselines may also distinguish different embodiments directly via their morphology observation. Consequently, PEAC's leading advantage may slightly decrease, but still significantly outperforms baselines (We have also supplemented experiments in DMC with varying morphologies in the global rebuttal). Thanks for your question, we will discuss it more in the revised versions.
>
> Reference:
>
> [1] URLB: Unsupervised Reinforcement Learning Benchmark
>
> [2] Behavior contrastive learning for unsupervised skill discovery
>
> [3] Robot Parkour Learning
>
> [4] Embodied intelligence via learning and evolution

---

> > ### Author Response · Authors · 2024-08-12
> > **Look forward to further feedback**
> >
> > Dear Reviewer 3BHi:
> >
> > We sincerely thank you again for your valuable and constructive comments. We have tried our best to answer the concerns raised, especially including more ablation studies and more complex tasks, and are happy to clarify/discuss any further questions. We hope you may find our response satisfactory and raise your rating accordingly. We are looking forward to hearing from you about any further feedback.
> >
> > Best, Authors.

---

> ### Comment · Area_Chair_YC1z · 2024-08-12
> **Reviewer Discussion Needed**
>
> Dear Reviewer,
>
> The discussion time is coming to an end soon. Please engage in the discussion process which is important to ensure a smooth and fruitful review process. Give notes on what parts of the reviewers responses that have and have not addressed your concerns.

---

> ### Author Response · Authors · 2024-08-14
> **Thank you again for your supportive comments at the end of the discussion**
>
> Dear Reviewer 3BHi:
>
> Thanks a lot for your supportive and constructive comments. As the rebuttal phase is nearing its end, we sincerely hope that you could take a moment to review our rebuttal. Your feedback is invaluable to us, and we truly appreciate your time and efforts in this review process. Thank you again for your time and effort in helping us improve our paper.
>
> Best, Authors.

---

### Author Rebuttal · Authors · 2024-08-06

We thank all the reviewers for their valuable comments which help to further improve our paper. Here we first address the common concerns on **baselines/ablation studies** and **more complicated cross-embodiment settings**. Then, we provide a detailed response to the comments of each reviewer respectively.

**Q1:** More baselines and ablation studies.

**A:** We appreciate the reviewer's recognition of the comprehensiveness of experiments (Reviewer RG3H) and the adequacy of baselines (Reviewer xr64). **To better highlight the contribution of this paper**, we have supplemented various ablation studies following the valuable suggestions of Reviewer 3BHi and RG3H.

First, we supplement **LBS-Context** and **DIAYN-Context**, i.e., combining LBS and DIAYN with the embodiment discriminator in PEAC, which utilizes embodiment information during the pre-training stage. Our results are as below:

| State-based DMC | Walker-mass | Quadruped-mass | Quadruped-damping |
|-|-|-|-|
| DIAYN  | 266.7  | 501.8| 416.7 |
| DIAYN-context  | 364.4  | 564.2 | 499.5  |
| LBS | 367.1 | 536.5 | 367.6 |
| LBS-context | 365.9 | 514.5  | 417.4|
| PEAC  | **463.7** | **643.6** | **572.9** |

| Image-based DMC | Walker-mass | Quadruped-mass | Quadruped-damping |
|-|-|-|-|
| DIAYN | 461.8 | 438.4  | 526.5  |
| DIAYN-context  | 572.0 | 457.6 | 546.6 |
| PEAC-DIAYN | 587.3 | 578.4 | 572.8 |
| LBS | 647.2 | 736.2 | 721.2 |
| LBS-context | 655.1  | 739.1 | 722.5  |
| PEAC-LBS | **727.6** | **755.2** | **740.2** |

**The figures with aggregate metrics of these methods are in the attached PDF**. As shown in these two tables and figures, LBS-Context and DIAYN-Context own superior performance compared with LBS and DIAYN respectively, and PEAC still significantly outperforms them. Especially, in state-based DMC, the improvement in the performance of LBS-Context and DIAYN-Context is more remarkable than in image-based DMC. It may be because, in image-based DMC, we take DreamerV2 as the RL backbone, which has encoded historical information for making the decision and is possible to distinguish different embodiments through historical information to some degree (Lines 210-216). Consequently, **this ablation study highlights that both the embodiment discriminator and cross-embodiment intrinsic rewards $\mathcal{R}_{\text{CE}}$ are effective for handling CEURL**.

Moreover, we supplement ablation studies of the hyperparameter $\beta$ in Eq.2 ($\beta$ is set to 1.0 in all our experiments), which is for balancing the policy improvement term and the policy constraint term.

| Ablation of $\beta$ | Walker-mass | Quadruped-mass |
|-|-|-|
| PEAC-LBS  | **727.6** | **755.2**  |
| PEAC-LBS($\beta$=0.5) | **726.9** | 728.8  |
| PEAC-LBS($\beta$=2.0) | 713.2 | **750.1**  |

As shown in this table, when $\beta$ is small, the performance of PEAC-LBS decreases a little, but PEAC-LBS is still stable with different $\beta$.

**Q2:** About experiments with more complicated cross-embodiment settings.

**A:** To the best of our knowledge, this is **the first work to consider unsupervised pre-training across different embodiments**. Consequently, when designing experimental settings, we hope to cover diverse kinds of cross-embodiment settings to fully validate algorithms in this setting, including different embodiment properties, different morphologies, different actions, and so on. In DMC, we mainly consider embodiments with different embodiment properties like mass and damping as this is **one of the most basic cross-embodiment settings** and is worth paying attention to especially when we first research CEURL. Besides these, there are also settings with **different morphologies** in **Appendix B.10**: **Walker-Cheetah**, of which the results show that PEAC can also handle embodiments with different morphologies.

Following reviewers' valuable suggestions, we have also supplemented more cross-embodiments with **different morphologies** in DMC:

- **Walker-Humanoid**: As suggested by Reviewer RG3H, we consider the cross-embodiment setting that includes Walker robots and Humanoid robots. Their robot properties, robot shapes, and action spaces are all different.
- **Walker-length** and **Cheetah-torsolength** [1]: The former includes walker robots with different foot lengths while the second one includes cheetah robots with different torso lengths. Thus robots' properties and morphologies are different. The figures of these embodiments are in the attached PDF.

The results in these settings are below:

| Walker-Humanoid | stand-stand | stand-walk | stand-run | walk-stand | walk-walk | walk-run | run-stand | run-walk | run-run | average      |
|-|-|-|-|-|-|-|-|-|-|-|
| DIAYN  | 286.8 | 320.2  | 307.2 | 185.4 | 175.4 | 209.7 | 75.4  | 61.1 | 55.4 | 186.3  |
| PEAC-DIAYN | 231.6 | 430.3  | 371.7  | 247.4 | 210.7 | 245.2 | 88.3 | 85.1 | 69.9  | 220.0  |
| LBS | 477.8 | 463.2   | 475.7 | **407.1** | 343.7 | 401.1 | 150.8  | 112.3 | 134.6 | 329.6  |
| PEAC-LBS | **486.4** | **484.9**  | **483.6**  | **406.9** | **373.6**  | **432.5** | **180.4** | **145.7** | **188.2** | **353.6**  |

| Walker-length | stand | walk | run| flip | average  |
|-|-|-|-|-|-|
| LBS | **977.5** | 909.1 | **557.5** | 630.4 | 768.6  |
| PEAC-LBS | **970.0**  | **976.4** | 544.2 | **764.9** | **813.9**  |

| Cheetah-torsolength | run | run_backward | flip  | flip_backward | average  |
|-|-|-|-|-|-|
| LBS | 625.9 | **512.9** | 611.9 | 523.0 | 568.4  |
| PEAC-LBS| **745.3** | 499.7 | **646.4** | **649.2** | **635.2** |

As shown in these tables, PEAC can achieve much greater performance compared to baselines. These experiments indicate that PEAC has powerful abilities to handle various kinds of embodiment differences, including different morphologies. Also, designing more effective methods for handling complicated embodiments like Humanoid is a promising future direction.

Reference:

[1] Learning Robust State Abstractions for Hidden-Parameter Block MDPs

---

### Decision · Program_Chairs · 2024-09-25

**Decision:**

Accept (poster)

**Comment:**

This work studies an area of cross-embodiment unsupervised RL, which is of growing interest in the community. The pros outweigh the cons for this paper and the analysis and discussion of the problem space is a fair contribution to the community. We encourage the authors to continue to consider the comments from the reviewers to improve the final version of the work.